# Paired yeast one-hybrid assays to detect DNA-binding cooperativity and antagonism across transcription factors

Anna Berenson [1], Ryan Lane [1], Luis F. Soto-Ugaldi[2], Mahir Patel [3], Cosmin Ciausu[3], Zhaorong Li[1], Yilin Chen[1], Sakshi Shah[1], Clarissa Santoso[1], Xing Liu[1], Kerstin Spirohn [4,5,6], Tong Hao [4,5,6], David E. Hill [4,5,6], Marc Vidal[4,5,6] & Juan I. Fuxman Bass [1,4] ✉

Cooperativity and antagonism between transcription factors (TFs) can drastically modify their binding to regulatory DNA elements. While mapping these relationships between TFs is important for understanding their context-specific functions, existing approaches either rely on DNA binding motif predictions, interrogate one TF at a time, or study individual TFs in parallel. Here, we introduce paired yeast one-hybrid (pY1H) assays to detect cooperativity and antagonism across hundreds of TF-pairs at DNA regions of interest. We provide evidence that a wide variety of TFs are subject to modulation by other TFs in a DNA region-specific manner. We also demonstrate that TF-TF relationships are often affected by alternative isoform usage and identify cooperativity and antagonism between human TFs and viral proteins from human papillomaviruses, Epstein-Barr virus, and other viruses. Altogether, pY1H assays provide a broadly applicable framework to study how different functional relationships affect protein occupancy at regulatory DNA regions.

Gene expression is controlled by the binding of transcription factors (TFs) to regulatory DNA elements to direct the recruitment of cofactors and the transcriptional machinery. The logic of transcriptional regulation by TFs is complex as some TFs can positively or negatively affect one another's ability to bind DNA[1–3]. This results in the binding of different combinations of TFs at promoters and enhancers, fine-tuning transcriptional output[4]. Some TFs bind DNA cooperatively, either via mutual cooperativity (e.g., as heterodimers or by indirect cooperativity mediated by DNA[5]), or when a DNA-bound TF recruits a second TF. Other TFs antagonize one another by sequestration via protein-protein interactions (PPIs) or by competing for binding at specific DNA sites (e.g., paralogs that recognize the same motif[6,7]). As a result of these functional relationships, individual TFs are often limited to binding DNA under certain conditions, such as in the presence of a cooperator or the absence of an antagonist.

Understanding these functional relationships between TFs at regulatory DNA regions is essential for mapping their roles in different contexts but has thus far been difficult to achieve experimentally. DNA binding predictions based on motif analysis often identify many more potential binding events than are observed in vivo[8,9]. Predictions are generally more challenging for TF heterodimers, as binding motifs have not been determined for most heterodimers due to challenges in producing and purifying protein complexes in vitro[10,11]. Single-molecule footprinting can be used to narrow down potential sites of co-binding of most TFs genome-wide; however, this approach still relies on the quality and availability of known DNA binding motifs, as well as their ability to predict TF dimer binding[12,13]. Other genome-wide

[1]Department of Biology, Boston University, Boston, MA 02215, USA. [2]Tri-Institutional Program in Computational Biology and Medicine, New York, NY, USA. [3]Department of Computer Science, Boston University, Boston, MA 02215, USA. [4]Center for Cancer Systems Biology (CCSB), Dana-Farber Cancer Institute, Boston, MA 02215, USA. [5]Department of Cancer Biology, Dana-Farber Cancer Institute, Boston, MA 02215, USA. [6]Department of Genetics, Blavatnik Institute, Harvard Medical School, Boston, MA 02115, USA. ✉e-mail: fuxman@bu.edu

experimental methods such as ChIP-seq[14] and CUT&RUN[15] profile one TF at a time. Therefore, cooperativity between TF-pairs is often inferred from correlation in binding profiles or determined using genetic perturbations (e.g., TF overexpression, knockout, or knockdown)[3,16,17]. Additionally, genome-wide experiments are limited to detecting interactions occurring in the cell types and conditions studied which could be influenced by local chromatin states and co-expression of multiple other TFs, obscuring functional relationships between TF-pairs of interest. Furthermore, these approaches typically focus on cooperative DNA binding but do not account for antagonistic relationships.

Enhanced yeast one-hybrid (eY1H) assays provide a complementary approach by mapping protein-DNA interactions (PDIs) on a TF-wide scale using a reporter-based readout[18–21]. eY1H assays evaluate interactions between an array of hundreds of TFs and different DNA regions of interest (e.g., promoters and enhancers) which are integrated into specific loci in the yeast genome. This allows the identification of the repertoire of possible PDIs at these DNA regions rather than binding events occurring in a specific condition or cell type. However, as each arrayed yeast strain only expresses one TF, eY1H assays typically cannot identify heterodimer-DNA interactions or other cooperative or antagonistic relationships between TFs[22].

Here, we introduce paired yeast one-hybrid (pY1H) assays, an adaptation of eY1H assays using TF-pair yeast arrays to detect cooperative binding and antagonism between hundreds of TF-pairs at DNA regions of interest. This approach reveals that these functional relationships occur across well-known and lesser-known TF-pairs in a DNA region-specific manner. Cooperative TF-pairs have significant evidence of in vivo co-binding in ChIP-seq experiments and often involve one ubiquitously expressed TF and one tissue-specific TF, while antagonistic pairs frequently involve two ubiquitous TFs. We also observe that different TF isoforms have varying functional relationships with other TFs, further expanding the TF interactome landscape. Furthermore, we show that viral proteins can antagonize the binding of human TFs to their DNA targets or direct them to new targets, providing mechanistic insight into host transcriptional reprogramming by viruses. Overall, pY1H assays constitute a robust and versatile approach to study functional relationships that modulate DNA targeting by TFs.

## Results

### pY1H assay design

eY1H assays utilize a DNA-bait yeast strain containing a DNA region of interest integrated into the yeast genome upstream of two reporter genes (HIS3 and lacZ) and a TF-prey strain expressing a TF fused to the Gal4 activation domain (AD). The DNA-bait and TF-prey yeast strains are mated pairwise using a robotic platform[18,23]. In the event of TF-DNA binding, the AD promotes the expression of both HIS3 (allowing yeast to overcome inhibition by the His3p competitive inhibitor 3-amino-1,2,4-triazole) and lacZ (producing a blue compound in the presence of X-gal), regardless of the intrinsic transcriptional activity of the TF. In pY1H assays, each TF-pair yeast strain expresses two TFs of interest, one or both of which are fused to an AD (Fig. 1a). The two TFs are cloned into different expression vectors (pAD2μ-TRP1 and pGADT7-GW-LEU2) to allow for selection using both the TRP1 and LEU2 markers. These vectors both have a 2 μ origin of replication and use the ADH1 promoters to express both TFs at similar levels, as evidenced by similar reporter activities for the same TF when expressed from each vector (Supplementary Fig. 1a). Reporter signal from the TF-pair yeast is compared to that from two corresponding single-TF control strains to detect reporter activation that is synergistic (i.e., the activity of the TF-pair is much stronger than either single-TF) or antagonistic (i.e., the activity of the TF-pair is much weaker than the activity of one of the single-TFs) (Fig. 1b). This system of event calling is supported by two main findings. First, it was previously observed that eY1H reporter signal strength correlates with signal from more quantifiable binding

reporter assays in mammalian cells[19]. Second, >90% of events detected in initial pY1H assays corresponded to obligate cooperative binding (where neither TF has any reporter signal in the absence of its partner) and complete antagonism (where a single-TF signal is completely lost in the TF-pair strain), minimizing reliance on signal strength comparisons. To analyze the pY1H data, we developed DISHA (Detection of Interactions Software for High-throughput Analyses), a computational pipeline and visual analysis tool for assessing reporter intensity and comparing yeast strains (Supplementary Figs. 2 and 3). By integrating DISHA analysis with manual curation, we identified cooperative and antagonistic events with a high level of reproducibility (Supplementary Fig. 1b).

We focused on two possible pY1H assay designs, the 1-AD design in which only one TF in each TF-pair is fused to an AD and the 2-AD design in which both TFs are fused to an AD. These assay designs can be applied to identify different types of functional relationships (Fig. 1c). By testing both possible AD orientations for each TF-pair (TF1-AD + TF2, TF1 + TF2-AD), the 1-AD design can be used to differentiate between two classes of cooperativity—mutual cooperativity and recruitment of one TF by another—and between two classes of antagonism—sequestration and competition. The 2-AD design can detect mutual cooperativity and sequestration using only one yeast strain per TF-pair, but cannot differentiate recruitment and competition from independent TF binding (Fig. 1c).

### Mapping relationships between NF-κB and AP-1 TF-pairs

NF-κB and AP-1 TFs often bind DNA as heterodimers, constituting a well-established model to benchmark pY1H assays and compare the 1-AD and 2-AD designs[24,25]. We evaluated the binding of 6 NF-κB and 21 AP-1 TF-pairs to the promoters of 18 cytokine genes, each known to be regulated by at least one NF-κB and one AP-1 subunit[26] (see Supplementary Tables 1–3 within the Supplementary Data file). By assessing results from the 1-AD design, we observed examples of mutual cooperativity, recruitment, sequestration, and competition, while the 2-AD design showed robust evidence of mutual cooperativity and sequestration, confirming the expected divergent uses of the two assay designs (Supplementary Fig. 4a). Interestingly, though sequestration is generally expected to cause global loss of binding of the sequestered TF, some sequestering relationships such as that between REL and RELB were DNA bait-specific, as RELB did not prevent REL binding at all promoters tested (Supplementary Fig. 4b). This suggests a mechanism in which TF dimerization forms a complex that retains DNA binding ability but has altered sequence specificity, as has been previously reported[27–29].

For further analysis, we considered the union of all cooperative events (including mutual cooperativity and recruitment) and antagonistic events (including sequestration and competition) observed using either assay design (See Supplementary Table 4 within the Supplementary Data file). Overall, we detected 40 cooperative binding events between 17 TF-pairs and 9 cytokine promoters (Fig. 1d–f). For 70% of these events, one or both TFs were known to bind the regulatory regions or regulate the expression of that cytokine, as per the CytReg Database (https://cytreg.bu.edu/search_v2.html)[26] (Fig. 1g). This suggests that pY1H assays can recapitulate known PDIs while revealing previously undetected interactions that require cooperativity, including 71 individual PDIs that were tested previously by eY1H and had shown no binding signal. Cooperative events identified using the two assay designs showed similar overlap with existing literature (Fig. 2g). We also observed 32 antagonistic events between 12 TF-pairs at 8 cytokine promoters (Fig. 1d–f). This includes antagonism of REL by RELB at 4 cytokine promoters (Fig. 1e), consistent with findings that RELB/RELB and REL/RELB dimers display reduced DNA binding compared to other NF-κB dimers[30,31], as well as previously unreported antagonistic AP-1 TF-pairs (Fig. 1f). Overall, this screen detected additional instances of DNA bait-specific cooperativity and antagonism

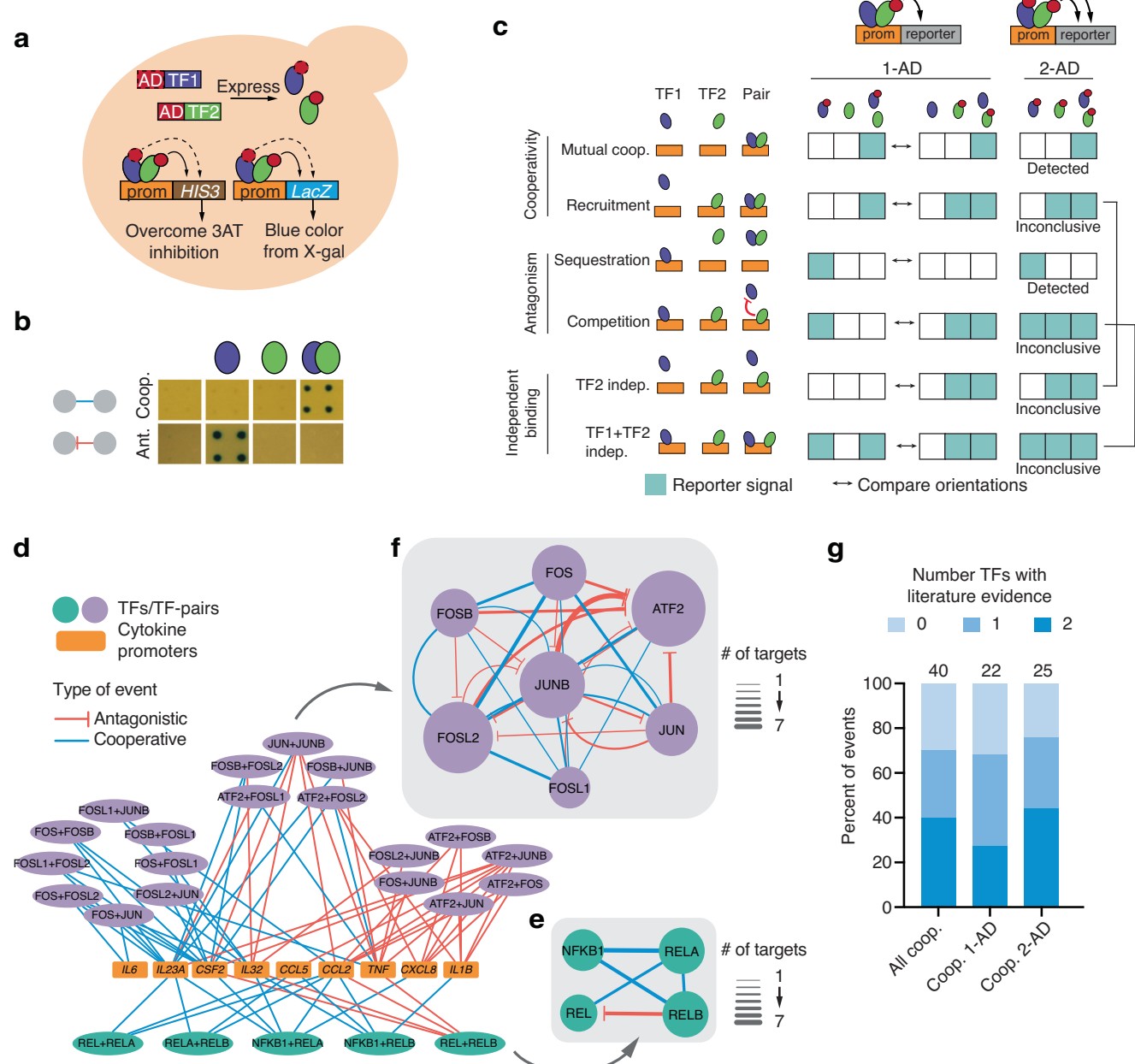

**Fig. 1 | Paired yeast one-hybrid (pY1H) assays. a** Schematic of pY1H assays. A DNA-bait yeast strain with a DNA sequence of interest (e.g., a promoter) cloned upstream of the *HIS3* and *lacZ* reporter genes is mated with a TF-pair prey strain expressing two TFs fused or not to the Gal4 activation domain (AD). If an AD-containing TF binds the DNA region of interest, reporter expression will allow the yeast to grow in media lacking histidine and in the presence of the His3p inhibitor 3-amino-1,2,4-triazole (3AT), and turn blue in the presence of X-gal. **b** pY1H assays detect cooperative and antagonistic interactions by comparing single-TF and TF-pair yeast strains. **c** Comparison between 1-AD and 2-AD screen designs for different cooperative (mutual cooperativity and recruitment), antagonistic (sequestration and competition), and independent DNA binding modalities. Teal boxes indicate cases

where reporter activity is expected. While the 1-AD design can distinguish between the six indicated binding modalities if reciprocal AD orientations are tested, the 2-AD design can only detect mutual cooperativity and sequestration. **d**–**f** Results of pY1H screen between NF-κB and AP-1 TF-pairs and cytokine gene promoters. **d** Main network shows connections between TF-pairs and cytokine promoters. **e**, **f** Cooperative and antagonistic relationships between NF-κB (**e**) and AP-1 (**f**) TFs. Node size indicates the number of binding events for that TF. Edge width represents the number of cooperative or antagonistic events involving a specific TF-pair. **g** Overlap of NF-κB and AP-1 pY1H interactions with the literature. Numbers above each bar reflect the number of binding events assessed in each category. Source data are provided as a Source Data file.

between highly-studied NF-κB and AP-1 TFs. This demonstrates the utility of pY1H assays to map these functional relationships and provides new information about how NF-κB and AP-1 subunits combine to enhance or inhibit targeting of certain promoters. Additionally, we observed the expected differences between the 1-AD and 2-AD assay designs, confirming their applicability to study different types of cooperative and antagonistic events.

## pY1H screen using a large-scale TF-pair array
We expanded the scope of pY1H assays by generating a large-scale TF-pair yeast array (Fig. 2a). We compiled a list of 868 TF-pairs based on reported PPIs or homology with interacting pairs (pTF1.0) (Fig. 2b, Supplementary Fig. 5a)[32,33] (see Supplementary Table 5 within the Supplementary Data file). We used TF-encoding ORF clones[34–36] (see Supplementary Table 6 within the Supplementary

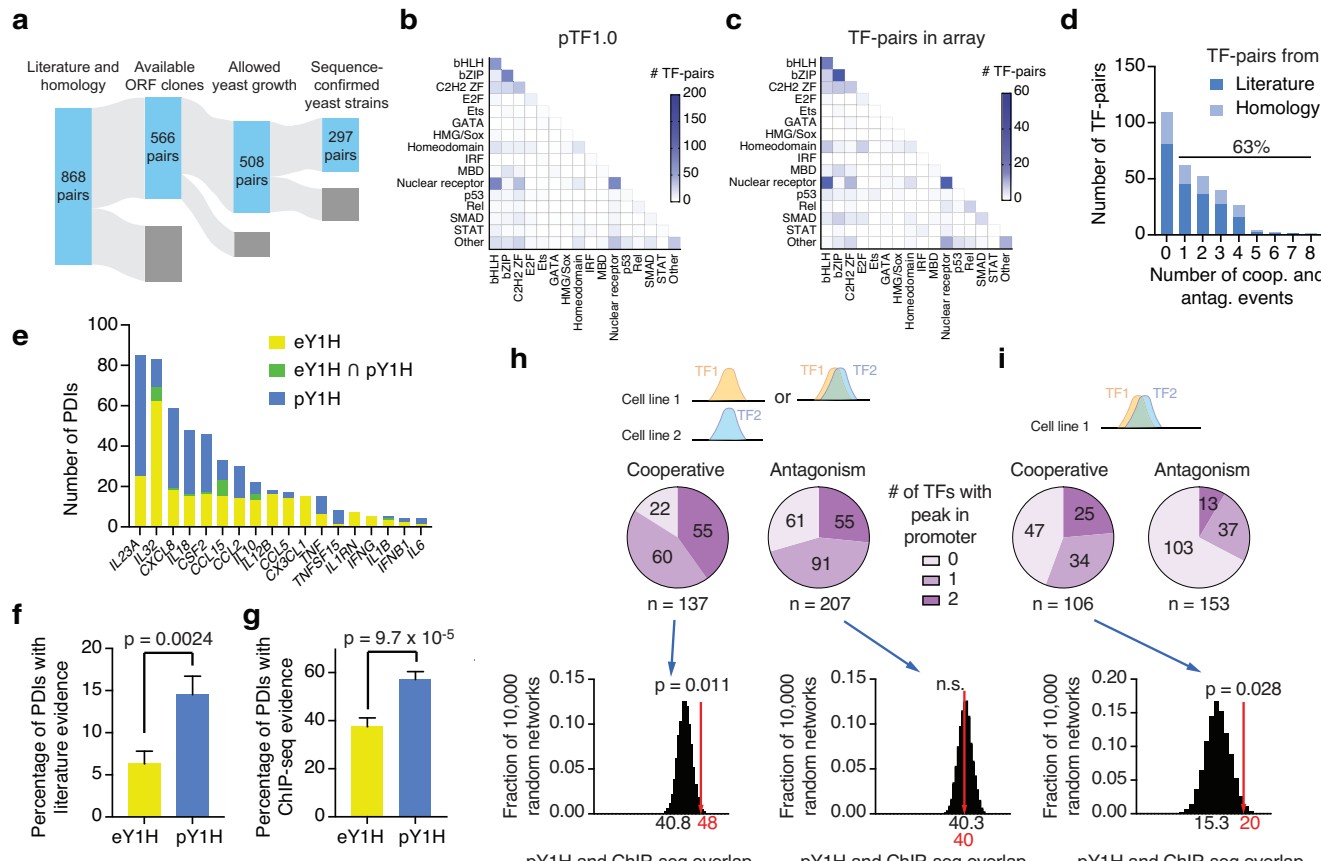

**Fig. 2 | Large-scale pY1H screen and validation. a** Generation of a large-scale TF-pair array for pY1H screening composed of 297 sequence-confirmed TF-pairs and their corresponding single-TF strains. **b**, **c** Number of TF-pairs for each TF family-pair in pTF1.0 (**b**) and in the TF-pair array (**c**). **d** Distribution of cooperative and antagonistic events detected for TF-pairs in our array. The percentage of TF-pairs with at least one cooperative or antagonistic event is indicated. **e** Comparison between eY1H protein-DNA interactions (PDIs) and cooperative PDIs by pY1H assays. **f** Percentage of eY1H ($n = 270$) and pY1H ($n = 256$) PDIs with literature evidence. Significance by two-tailed proportion comparison test. Error bars represent the standard error of proportion. **g** Percentage of eY1H ($n = 176$) and pY1H ($n = 226$)

PDIs with ChIP-seq evidence. Significance by two-tailed proportion comparison test. Error bars represent the standard error of proportion. **h, i** Comparison of pY1H results with ChIP-seq data from GTRD. For pY1H interactions, we indicate whether ChIP-seq peaks for one or both TFs have been reported in any cell line (**h**) and in the same cell line (**i**). Overlap between pY1H results and ChIP-seq peaks was compared to distributions of overlap for 10,000 randomized pY1H networks. Two-tailed statistical significance was calculated from Z-score values assuming normal distribution for overlap with the randomized networks. Source data are provided as a Source Data file.

Data file) to generate TF-prey yeast strains and sequence confirmed a final array of 297 TF-pairs (see Supplementary Table 7 within the Supplementary Data file), which has a similar distribution of TF families as pTF1.0 (Fig. 2c, Supplementary Fig. 5b). Given that the TF-pairs in our array are known or suspected to function as heterodimers, we selected the 2-AD assay design to robustly detect mutual cooperativity (hereafter "cooperativity") and sequestration (hereafter "antagonism") using a minimal number of yeast strains. We conducted a pY1H screen between these 297 TF-pairs and 18 cytokine promoters (see Supplementary Table 1 within the Supplementary Data file) and detected 180 cooperative binding events and 257 instances of binding antagonism across 15 cytokine promoters (see Supplementary Table 8 within the Supplementary Data file). Of the TF-pairs tested, 63% showed at least one cooperative or antagonistic interaction, including 60 of the 88 TF-pairs selected based on homology (Fig. 2d). Specifically, 32% of TF-pairs showed at least one cooperative interaction and 38% of TF-pairs showed at least one antagonistic interaction (Supplementary Fig. 6). These pairs involve TFs from a variety of families and include both intra- and inter-family TF-pairs (Supplementary Fig. 5c–f), suggesting that cooperative binding and antagonism are prevalent for a wide range of TF-pairs. From our cooperative binding events, pY1H assays revealed an additional 234 individual PDIs not previously detected by eY1H

assays at the cytokine promoters tested (Fig. 2e). Overlap between cooperative binding-derived PDIs and eY1H interactions is minimal, as eY1H cannot detect interactions that require cooperative binding and we excluded any independent binding events by individual TFs from our pY1H analysis. More importantly, when compared to eY1H PDIs, pY1H-derived PDIs showed a greater overlap with the literature (~6% vs. ~14% overlap, $p = 0.0024$ by two-tailed proportion comparison test) and with available ChIP-seq peaks (~38% vs ~57% overlap, $p = 9.7 \times 10^{-5}$ by two-tailed proportion comparison test) (Fig. 2f, g), demonstrating that pY1H assays can recover known PDIs not detectable by eY1H assays.

pY1H cooperative events significantly overlapped with motif predictions and ChIP-seq data (Fig. 2h, i and Supplementary Fig. 7). For 40% (55/137) of cooperative interactions with available data, both TFs have ChIP-seq peaks in the promoter in at least one cell line, a significantly greater overlap than expected for a randomized network (Fig. 2h). Furthermore, for cell lines with ChIP-seq data for both TFs, 24% (25/106) of cooperative interactions had ChIP-seq peaks for both TFs in the same cell line, which was also greater than expected for a randomized network (Fig. 2i). This provides strong evidence for in vivo co-binding of our cooperative TF-pairs at the target promoters identified. ChIP-seq overlap for antagonistic TF-pairs was not significant (Fig. 2h). This was expected, as we hypothesize that our antagonistic

events represent sequestration rather than competitive binding of both TFs.

## TF-TF relationships are DNA region-specific and connect ubiquitous and tissue-specific TFs

While 83 TFs participated exclusively in either cooperativity or antagonism across the cytokine promoters tested, 54 TFs, including FOS and others typically considered to be mainly cooperative, participated in both event types, suggesting that individual TFs have distinct functional relationships with different TF partners (Fig. 3a–c). Interestingly, 21 TF-pairs were cooperative or antagonistic depending on the promoter sequence (Fig. 3c), likely due to motif presence, spacing, and orientation. For example, MXI1 antagonized MAX at the *IL18* and *CCL15* promoters which have MAX motifs but no MXI1 motifs, while both TFs cooperated at the *CCL5* promoter that has overlapping MAX/MXI1 motifs at two locations (Supplementary Fig. 8a). The observed differences in functional relationships with TF partners even extend to paralogous TFs. While some sets of highly similar TF paralogs showed identical relationships with TF partners, others showed major differences in both their TF-TF relationships and DNA targets (Fig. 3d and Supplementary Fig. 8b). This suggests partner and target neofunctionalization and subfunctionalization between paralogs, and may explain the limited specificity observed for DNA binding predictions that rely on very similar motif preferences between paralogs.

Cooperativity and antagonism may be mechanisms by which tissue- and cell type-specific TFs modulate the function of more ubiquitous TFs. Using single-cell RNA-seq data from the Tabula Sapiens atlas[37], we calculated a tissue/cell type expression specificity score (TCESS) for TFs in pairs demonstrating cooperativity and/or antagonism, where TFs with TCESS ~ 1 are ubiquitously expressed and higher values indicate greater tissue specificity (see Supplementary Tables 9 and 10 within the Supplementary Data file). We observed that these functional relationships often occur between ubiquitous-ubiquitous and ubiquitous-specific TF-pairs (Fig. 3e). Even for ubiquitous-specific TF-pairs, TFs were expressed in overlapping sets of tissues, with 97% of all TF-pairs coexpressed in at least one tissue or cell type (Fig. 3f), indicating potential venues for cooperative and antagonistic interactions to occur in vivo. Interestingly, TFs in cooperative pairs had a significantly greater difference in TCESS than TFs in antagonistic pairs, while the expression overlap was similar for both types of TF-pairs (Fig. 3g). This suggests that cooperativity is the preferred mechanism for modulation of ubiquitous TFs by tissue-specific TFs, as cooperative events more commonly occur between ubiquitous-specific pairs, while antagonism may constitute a broader mechanism whereby pairs of ubiquitous TFs limit one another's DNA binding across a wide range of tissues and cell types.

## Identifying highly cooperative and frequently antagonized TFs

Cooperative binding events were observed between 95 TF-pairs from diverse TF families (Supplementary Fig. 5c, d). About 90% of these events indicated obligate cooperative binding, while about 10% showed enhanced binding of one or both TFs. This includes known heterodimers such as bHLH, nuclear hormone receptor, bZIP, and Rel pairs (Supplementary Fig. 5c, d). Interestingly, we observed many TFs that participated in a disproportionate number of cooperative binding events (e.g., TP53, RXRA, RELA, and IKZF3) many of which, to our knowledge, have not been reported. This confirms the utility of pY1H assays to identify cooperative events in an unbiased manner.

Extensive antagonism was also observed between 114 TF-pairs (Supplementary Fig. 5e, f). Some TFs such as NCOA1, FOS, MAX, and RARB were frequently antagonized (Fig. 3a), suggesting that these TFs are highly influenced by the repertoire of co-expressed TFs. While most TFs functioned exclusively as antagonists or antagonized TFs in our screen, 27 TFs participated in each role at different promoters, suggesting that the role of a given TF depends on its TF partner as well

as the target DNA sequence (Fig. 3h). This is likely due to differences in specificity between the individual TFs.

## Alternative isoform usage alters TF-TF relationships

Most human TFs are expressed as multiple isoforms, expanding the number of functionally distinct TFs[38,39]. We used pY1H assays to determine whether alternative isoforms of a given TF differ in their functional relationships with other TFs. We screened 37 TF isoform-pairs involving immune-related TFs for binding to 102 cytokine gene promoters (Fig. 4a) (see Supplementary Tables 1, 11, and 12 within the Supplementary Data file). Alternative isoforms often differed in binding modalities, in many cases switching between dependent binding types (cooperative and antagonistic) (Fig. 4a, b, see Supplementary Table 13 within the Supplementary Data file). For example, while the STAT1-202 isoform showed cooperative binding with IRF9, the STAT1-201 isoform antagonized IRF9 binding (Fig. 4c). In other cases, alternative isoforms had varying levels of dependence on other TFs, switching between dependent and independent binding. For example, DNA binding of the MAX-205 isoform was typically independent of MNT, while binding of the MAX-202 isoform was always antagonized by MNT (Fig. 4d).

Although the binding modalities were often similar across DNA targets for specific isoform-pairs, in other cases the effect of isoform usage differed between promoters. For PPARG/RXRG and RARG/RXRG, alternative isoforms showed identical binding modalities at some promoters (Fig. 4b green arrows) and divergent modalities at other promoters (Fig. 4b magenta arrows).

As alternative TF isoforms can differ in both DNA binding and PPIs due to gain or loss of different protein domains, we suspect that alternative isoform usage can affect DNA binding modalities by multiple different mechanisms. For example, STAT3-203 shows mostly cooperative binding with STAT1-202 but is antagonized by STAT1-212, a truncated isoform missing its DNA binding domain, suggesting that the STAT3/STAT1-212 dimer has reduced DNA binding affinity (Fig. 4a). However, STAT3 binding is also antagonized by the STAT1-201 isoform, which retains its DNA binding domain but has an additional C-terminal domain. To determine the potential mechanism of antagonism, we used Alphafold 2 to predict structures of dimers between STAT3 and the STAT1-202 and STAT1-201 isoforms. We observed that the additional C-terminal region in STAT1-201 likely does not interfere with STAT1-STAT3 dimerization in the antiparallel conformation (where the C-terminal domains are distal from the site of dimerization), but could interfere with dimerization in the parallel conformation, which is the primary conformation for DNA binding[40,41] (Supplementary Fig. 9). This supports an antagonistic mechanism by which STAT1-201 dimerizes with STAT3, decreases the number of STAT3 subunits available to form STAT3-STAT3 homodimers, and forms a STAT1-STAT3 dimer that is unable to bind DNA. Altogether, these findings suggest that alternative isoforms may affect DNA targeting by forming complexes with altered DNA binding specificity/affinity or due to differences in PPIs.

## Viral proteins alter DNA targeting of host genes by human TFs

Viruses express viral transcriptional regulators (vTRs) that can modulate host gene expression, altering immune responses, apoptosis, differentiation, and cell cycle dynamics[42]. vTRs participate in extensive interactions with human proteins[42–44], but less is known about the functional outcomes of these interactions. We leveraged pY1H assays to investigate mechanisms by which vTRs affect binding of human TFs to gene promoters (Fig. 5a). We generated a pY1H array of 113 protein pairs containing one human TF and one vTR that are known or suspected to interact by PPIs (Fig. 5b) and screened for interactions with 83 promoters of cancer-related genes (see Supplementary Tables 1, 14, and 15 within the Supplementary Data file). We observed both cooperativity (8 events) and antagonism (42 events) between 11 vTRs and 11

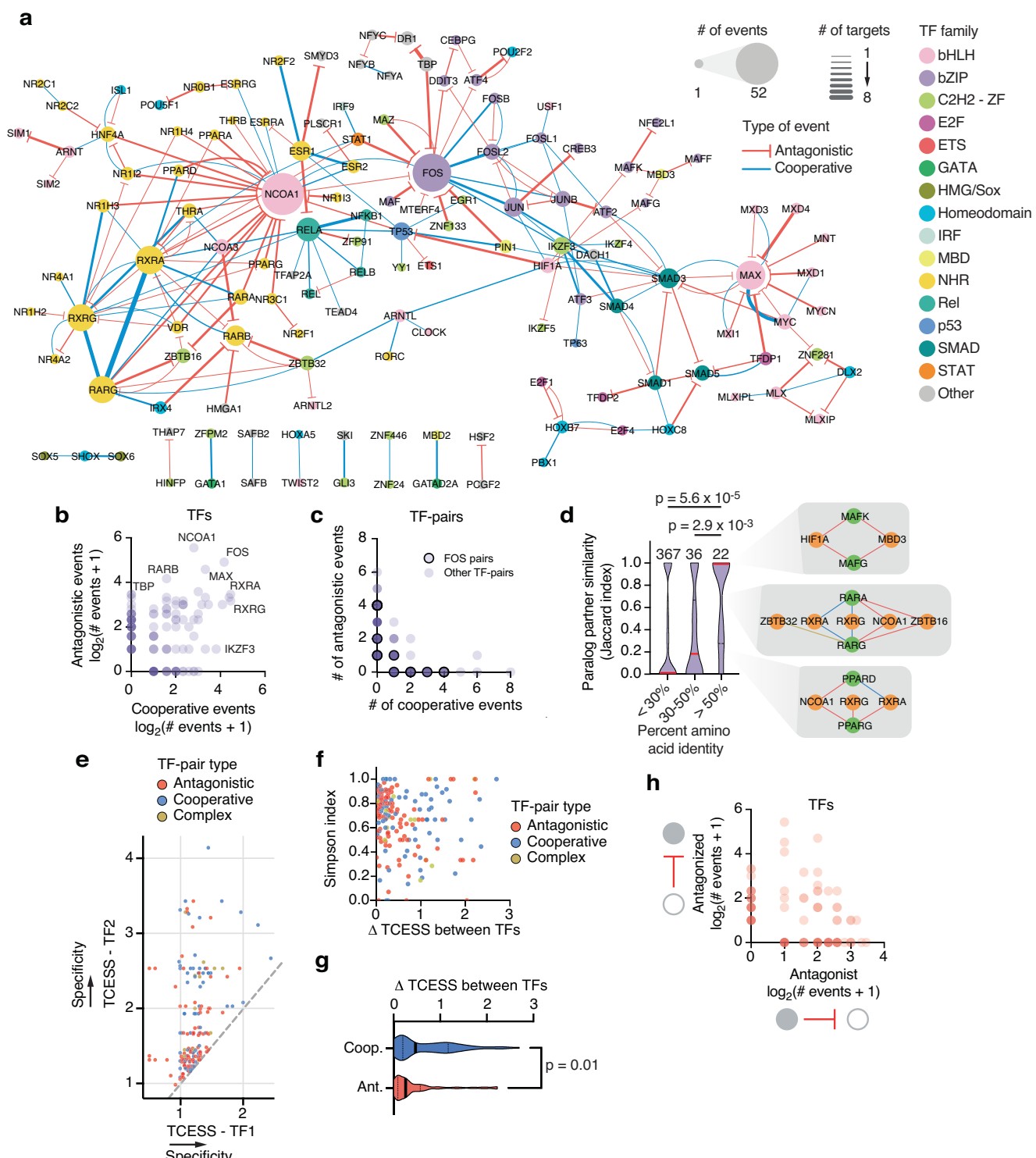

human TFs (Fig. 5c, see Supplementary Table 16 within the Supplementary Data file). Interestingly, the HBZ protein from human T-lymphotropic virus 1 (HTLV-1) cooperated with human DDIT3 to bind two promoters, but antagonized the binding of CEBPG to four promoters, although both DDIT3 and CEBPG are bZIP TFs. This indicates that a given vTR can have different effects on human TFs, even within the same TF family. Distinct vTRs from a virus can also have different effects on the binding of a human TF. For example, Epstein-Barr virus proteins EBNA3B and EBNA3C cooperated with and antagonized RBPJ, respectively, providing a potential mechanism for observations that EBNA3 proteins alter the expression of distinct sets of host genes via

interactions with RBPJ[45–47] (Fig. 5d). Most of the functional relationships we found between vTRs and human TFs were not previously reported and therefore provide evidence suggesting that different viruses can rewire host gene regulatory networks by altering host TF targets.

## Discussion

In this study, we introduce pY1H assays to identify DNA-binding cooperativity and antagonism across broad arrays of proteins, circumventing limitations often encountered by other approaches such as reliance on known DNA binding motifs, dependence on endogenous

**Fig. 3 | pY1H maps cooperative and antagonistic relationships between TFs.**
**a** Network of cooperative and antagonistic relationships between TFs at cytokine promoters screened. Node size indicates the number of binding events for that TF. Edge width represents the number of cooperative or antagonistic events involving a specific TF-pair. **b** Number of cooperative and antagonistic events observed for individual TFs. **c** Number of cooperative and antagonistic events observed for TF-pairs. FOS-containing pairs are outlined in black. **d** Similarity in cooperative and antagonistic relationships with shared TF partners (Jaccard index) between paralogs. Significance determined by two-tailed Mann–Whitney's *U*-test. Numbers above each column reflect the number of TF paralog pairs assessed in each group (*n* = 367 TF paralog pairs with <30% amino acid identity, *n* = 36 TF paralog pairs with 30%–50% amino acid identity, and *n* = 22 TF paralog pairs with >50% amino acid identity). Red dividing lines within each violin plot represent the median, and black

lines represent the top and bottom quartiles. Insets show relationships between paralog-pairs (green) with partners (orange). Edges in red, blue, and gold indicate antagonistic, cooperative, and complex relationships, respectively. **e** Tissue/cell-type expression specificity score (TCESS) for TFs in pairs showing cooperativity, antagonism, or both (complex). For each TF-pair, the larger TCESS value was plotted on the y-axis. **f** Scatter plot showing the Simpson co-expression similarity and the difference in TCESS for each TF-pair showing cooperativity, antagonism, or both (complex). **g** Difference in TCESS between TFs in cooperative TF-pairs (*n* = 72 pairs) and antagonistic TF-pairs (*n* = 93 pairs). Significance by two-tailed Mann–Whitney's *U*-test. Solid dividing lines within each violin plot represent the median, and dashed lines represent the top and bottom quartiles. **h** Number of antagonistic events in which each TF acted as the "antagonist TF" or "antagonized TF". Source data are provided as a Source Data file.

protein expression, and chromatin-related confounders. Studies of TF-TF relationships have primarily focused on cooperativity, namely in the context of heterodimer-DNA binding[16,48,49]. However, our work shows that DNA binding antagonism between TFs is equally common and may play an equivalent role in conveying regulatory specificity. Additionally, we observed that both cooperativity and antagonism extend to a wide range of TFs, many of which were not previously thought to function as heterodimers, highlighting the need for TF-wide approaches to identify these types of functional relationships.

Our results also show that DNA binding of a TF depends heavily on the repertoire of TFs and other proteins in the nucleus. While numerous studies have explored the effect of chromatin states on TF binding[50–52], our findings suggest that TF-TF relationships may also contribute to the drastic differences in genome-wide binding patterns of TFs observed across tissues and cell types, and help explain the limited expression correlation often observed between TFs and their target genes[53]. Additionally, we found that isoform variants and viral proteins drastically alter DNA targeting by TFs, which may contribute to differences in TF function across tissues and in certain disease states (e.g., in cancers that alter splicing patterns or during viral infection). Integrating TF-TF relationships observed by pY1H assays with genome-wide mapping of TF-DNA binding in different cellular contexts may better inform machine learning efforts to predict enhancer and promoter activity based on sequence and provide mechanistic insights into gene dysregulation in disease.

pY1H assays identify cooperative and antagonistic interactions in a heterologous context by expressing two TFs at a time. Therefore, orthogonal experiments may be required to determine the specific contexts in which these events occur, or whether they are affected by post-translational modifications (e.g., IRFs and STATs[54]) or by one TF targeting the other for degradation (e.g., viral HPV-16 E7[55,56]). However, using a heterologous assay has the advantage of interrogating the direct effects of DNA sequence on binding patterns of TF-pairs in the absence of other TFs from the same species that could have confounding interactions with the TFs evaluated.

pY1H assays can be used for diverse applications, leveraging both the 1-AD and the 2-AD designs. While the 1-AD design can be used to distinguish between a greater number of distinct binding modes and is likely to capture more dependent binding events, the 2-AD design efficiently detects mutual cooperativity and sequestration, two key mechanisms by which TFs affect one another's DNA occupancy. An immediate advance for this approach would involve expanding the human TF-pair array to incorporate all known and predicted TF-pairs. Pairs of isoforms or mutants of the same TF can also be studied to detect potential functional switches or dominant negative effects between them. pY1H assays can also be applied to study the binding and functional relationships between TFs from non-human species, leveraging existing Gateway-compatible TF clone resources from *Caenorhabditis elegans*[18], *Drosophila melanogaster*[20], *Mus musculus*[57], and *Arabidopsis thaliana*[21]. Additionally, pY1H assays can be used to study interactions involving other proteins within the nucleus,

including cofactor or scaffold protein recruitment by TFs, as well as expanded arrays of viral/human and viral/viral protein pairs. In summary, pY1H assays provide widespread evidence of complex functional relationships between TFs and constitute a broadly applicable method for studying occupancy of protein pairs at DNA regions of interest.

## Methods
### Ethical Statement
This research complies with all relevant ethical regulations and was approved by the Boston University Institutional Biosafety Committee under protocol #2211.

### TF-pair and DNA-bait selection
For our initial pY1H screen, we selected all 6 possible pairs of available NF-κB clones (NFKB1, REL, RELA, and RELB) and all 21 possible pairs of available AP-1 clones (FOS, FOSB, FOSL1, FOSL2, JUN, JUNB, ATF2). Of these 27 pairs, 24 were tested using both the 1-AD and 2-AD screen designs, and 3 were tested only in the 1-AD design (see Supplementary Table 3 within the Supplementary Data file). Using the CytReg2.0 database[26], we selected 18 cytokines that have been shown to be regulated by at least one NF-κB subunit and at least one AP-1 subunit (see Supplementary Table 1 within the Supplementary Data file). Yeast DNA-bait strains corresponding to the promoters of these cytokines (which were previously generated[26]) were screened against the collection of NF-κB and AP-1 TF-pairs and single-TFs.

For the large-scale TF-pair array, we selected all 429 TF-pairs with PPIs reported in the LitBM database[32]. We then added all 252 additional TF-pairs with more than two pieces of PPI evidence in the BioGRID database[33]. Finally, we added 187 pairs based on amino acid identity with selected pairs (See "Predicting possible TF-TF interactions based on homology" below). This resulted in an initial list of 868 TF-pairs, which we named pTF1.0 (see Supplementary Table 5 within the Supplementary Data file). After cloning, yeast transformations, and sequence confirmation, we obtained a final array of 297 TF-pairs for screening (see Supplementary Table 7 within the Supplementary Data file). We selected the same 18 cytokine promoters tested in the initial screen to use as DNA-baits (see Supplementary Table 1 within the Supplementary Data file).

To study alternative isoforms, we selected TFs with known immune regulatory functions: FOS, MAX, STAT1, STAT3, PPARG, RARG, and RXRG. We studied isoforms for these TFs available from the TFIso1.0 collection from the Center for Cancer Systems Biology (CCSB) at the Dana-Farber Cancer Institute and included a subset of TF partners for these TFs from the TF-pair array. This resulted in a final array of 37 TF isoform-pairs for screening (see Supplementary Table 12 within the Supplementary Data file) against 119 cytokine promoters for which DNA-bait yeast strains were previously generated[26] (see Supplementary Table 1 within the Supplementary Data file).

To determine cooperativity and antagonism between viral transcriptional regulators (vTRs) and human TFs, we used VirHostNet[43], Uniprot, and primary literature to select pairs of vTRs and human TFs

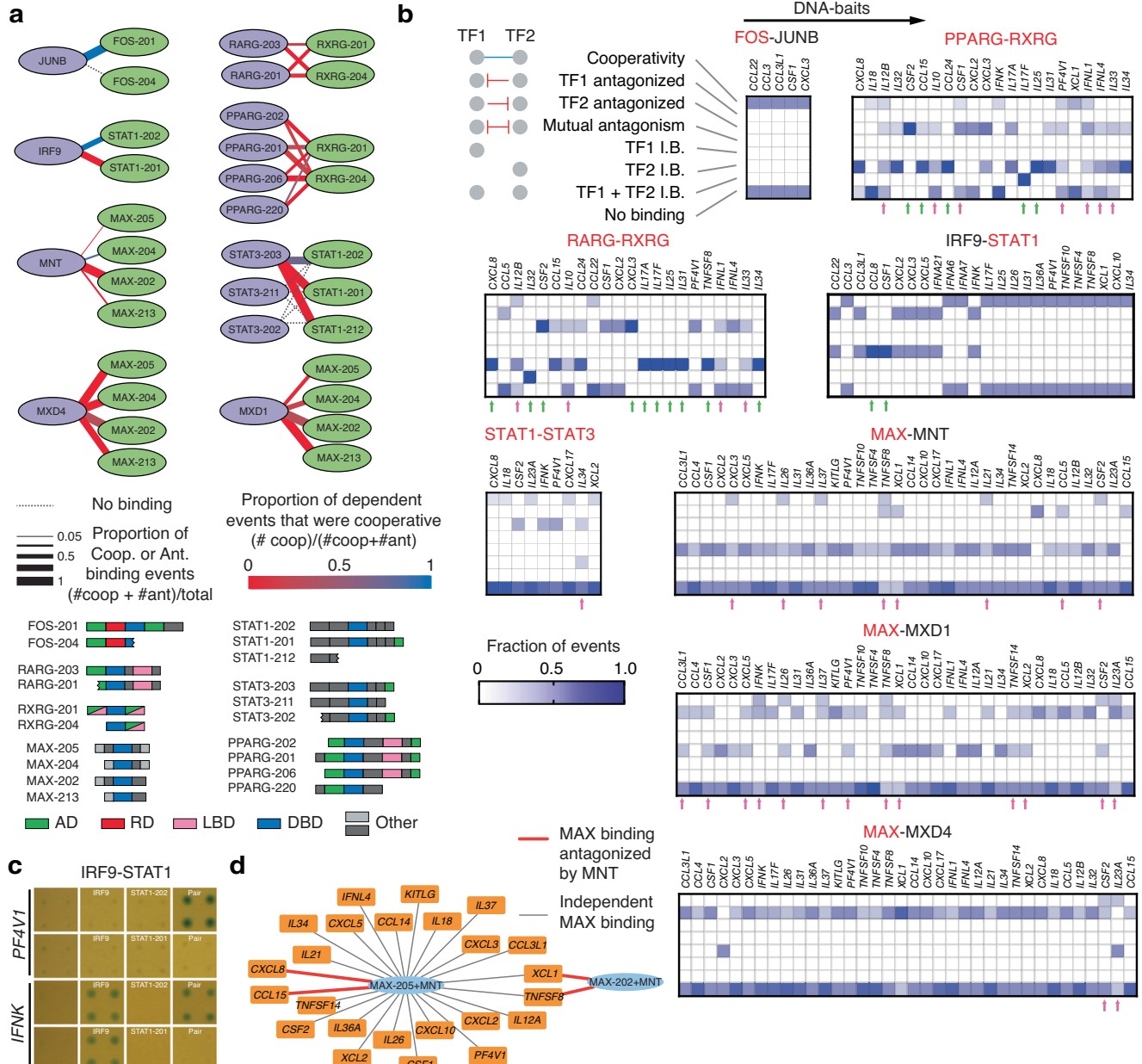

**Fig. 4 | Application of pY1H to study TF isoforms. a** Relationships between TF isoform-pairs observed by pY1H assays. Edge width represents the proportion of binding events for each TF-pair that were dependent (i.e., cooperative or antagonistic). Edge color represents the proportion of dependent events that were cooperative. Domain-based schematics of TF isoforms studied are shown. **b** For each TF-pair, the proportion of TF isoform-pairs that show each type of binding modality (cooperativity, TF1 or TF2 antagonized, mutual antagonism, TF1 and/or TF2 independent binding, or no binding) across DNA-baits. Names in red indicate TFs for which alternative isoforms were studied. Green arrows indicate DNA-baits

where all TF isoform-pairs for a given TF-pair show identical binding modalities; magenta arrows indicate DNA-baits where different TF isoform-pairs for a given TF-pair show at least three different binding modalities. **c** Relationship between alternative STAT1 isoforms and IRF9 at the *PF4V1* and *IFNK* promoters.
**d** Interactions between MAX-MNT dimers and cytokine promoters. Gray lines indicate independent MAX binding to the cytokine promoter, whereas red lines indicate that MAX binding was antagonized by MNT. Source data are provided as a Source Data file.

which have been shown to interact via PPIs. We supplemented these with additional vTR-TF pairs based on homology with known pairs to include similar proteins across viruses (e.g., E7 from HPV-2 and E7 from HPV-5). Once filtered for available ORF clones, this resulted in an initial list of 353 protein pairs. After cloning, yeast transformations, and sequence confirmation, we generated a final array of 113 vTR-TF pairs for screening (see Supplementary Table 15 within the Supplementary Data file). For DNA-baits, we selected 83 promoters of genes associated with cancer (see Supplementary Table 1 within the Supplementary Data file).

**Predicting possible TF-TF interactions based on homology**

PPIs involving human TFs were downloaded from the LitBM database[36]. For all analyses, we considered all 1639 human TFs reported in the Lambert list[58]. To identify possible TF-TF interactions, we used the following approach:

1. If two TFs ($TF_x$ and $TF_y$) were reported to interact in LitBM; then, each $TF_a$ highly similar to $TF_x$, and each $TF_b$ highly similar to $TF_y$ was considered as new possible pairs of interactors ($TF_x$ and $TF_b$, $TF_a$ and $TF_y$, and $TF_a$ and $TF_b$).

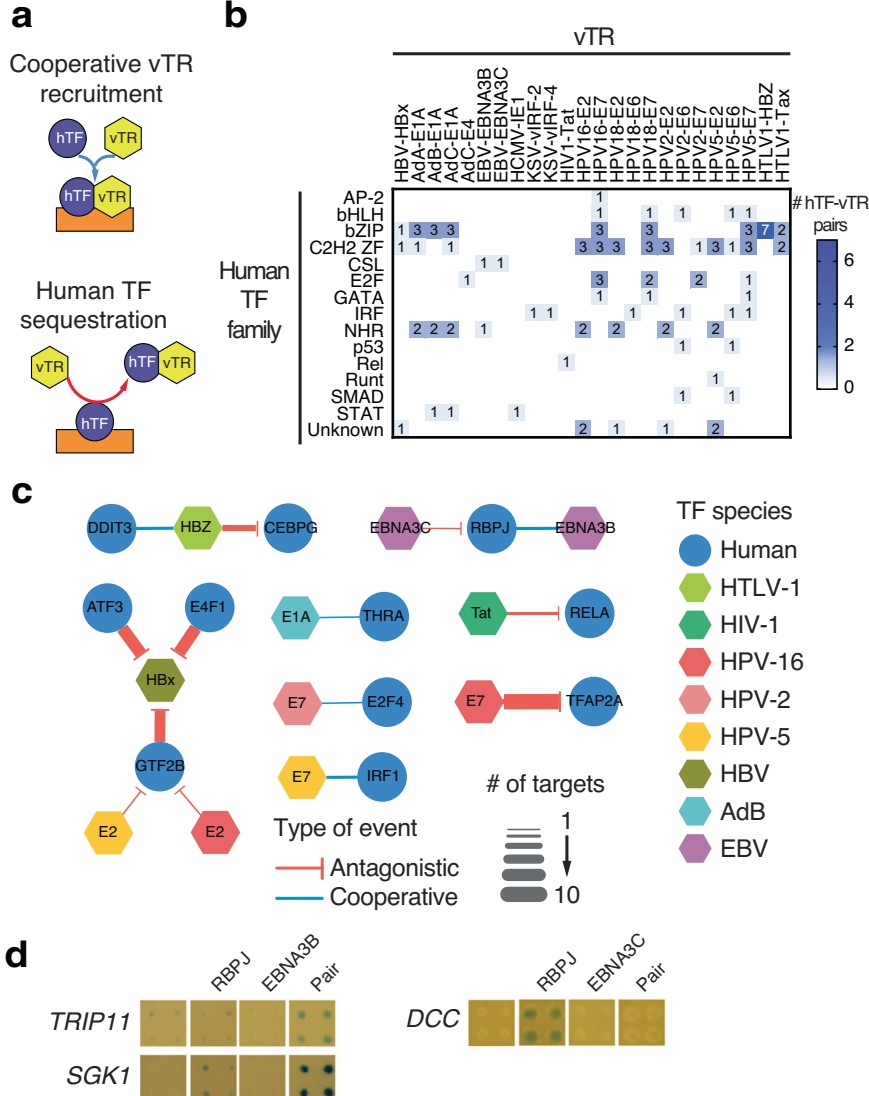

**Fig. 5 | Application of pY1H to study viral transcriptional regulators (vTRs).**
**a** Examples of models by which vTRs can affect human TF (hTF) binding. vTRs can cooperate with hTFs to bind to DNA elements or a vTR can sequester an hTF, preventing its binding to DNA. **b** Number of hTF-vTR pairs tested for binding to 83 cancer gene promoters. hTF-vTR pairs were selected based on known PPIs between the two proteins or homology with known pairs. hTFs are classified by TF families. **c** Network of relationships between hTFs and vTRs at 83 cancer gene promoters. **d** Examples of RBPJ-EBNA3B cooperative binding to the *TRIP11* and *SGK1* promoters, and of EBNA3C antagonism of RBPJ binding to the *DCC* promoter. Source data are provided as a Source Data file.

2. To determine the amino acid sequence similarity between TFs, the percent identity was determined using multiple alignments performed using Clustal 2.1[59]. A cutoff of 68.83% was used to identify highly similar TFs, as this corresponds to the 99.9th percentile in the percent identity matrix.

Code for this analysis can be found in the section "Predicting possible TF-TF interactions based on homology" within https://github.com/jfuxman/PY1H_NatComm2023/.

**Generation of TF-pair prey background yeast strain**
pY1H assays require transformation with two TF-prey plasmids. We selected the *TRP1* and *LEU2* as selection markers for these plasmids. Given that the Yα1867 yeast strain used for eY1H assay is *TRP1*- but *LEU2*+, we disrupted the endogenous *LEU2* gene in Yα1867 yeast using the M3926 leu2::KanMX3 disruptor converter plasmid with G418 resistance (Addgene #51680). M3926 was digested with BamHI (New England Biolabs R3136S) and ethanol precipitated.

Yα1867 yeast were transformed with digested plasmid as follows. Yeast were inoculated in 1 L liquid YAPD media to a concentration of OD600 = 0.15 and were incubated at 30 °C shaking at 200 rpm until they reached OD600 = 0.5, washed with sterile water, and washed again with 1X TE + 0.1 M lithium acetate (TE/LiAc). Yeast were then resuspended in TE/LiAc with salmon sperm DNA (ThermoFisher 15632011) at a dilution of 1:10 before adding 2 μg digested plasmid. Six volumes of TE/LiAc + 40% polyethylene glycol were added and samples were mixed gently ten times. Yeast were incubated at 30 °C without shaking for 30 min followed by 42 °C for 20 min, then resuspended in sterile water and plated on YAPD-agar with 100 μg/mL G418 (GoldBio G-418-1). We confirmed that Yα1867Δ*leu2* yeast were unable to grow on media lacking leucine.

**Generation of TF-pair ORF collections and yeast strains**
Most human TF ORFs were obtained from ORFeome 8 and 9 collections from the CCSB[32,34–36], while the remaining TF ORFs were obtained from the eY1H human TF ORF collection[60] (see Supplementary Tables 2

and 6 within the Supplementary Data file). Alternative TF isoform clones were obtained from the TFIso1.0 collection from the CCSB (see Supplementary Table 11 within the Supplementary Data file). vTR ORF clones were synthesized by GeneArt (see Supplementary Table 14 within the Supplementary Data file). All clones were obtained as Gateway Cloning-compatible entry clones and transferred to the corresponding destination vectors by LR cloning.

TF ORFs were cloned into yeast expression vectors using LR Gateway Cloning (ThermoFisher #11791100). For each TF-pair, one TF was cloned into the pAD2μ-*TRP1* (Walhout lab) plasmid and the other TF was cloned into the pGADT7-GW-*LEU2* plasmid (Addgene #61702).

Cloned TF-pairs were transformed into Yα1867Δ*leu2* yeast simultaneously, as previously described[60] and as follows. Yeast were inoculated in 1 L liquid YAPD media to a concentration of OD600 = 0.15 and were then incubated at 30 °C shaking at 200 rpm until they reached OD600 = 0.5, washed with sterile water, and washed again with 1X TE + 0.1 M lithium acetate (TE/LiAc). Yeast were resuspended in TE/LiAc with salmon sperm DNA (ThermoFisher #15632011) at a dilution of 1:10 before adding ~250 ng of each TF clone. Six volumes of TE/LiAc + 40% polyethylene glycol were then added and samples were mixed gently ten times. Yeast were incubated at 30 °C without shaking for 30 min followed by 42 °C for 20 min, then resuspended in sterile water. Transformed yeast were plated on selective media lacking tryptophan and leucine to select for double transformants.

All clones and yeast strains are available upon request made to corresponding author J.I.F.B., and will be shipped within 1 month of request.

## Generation of DNA-bait yeast strains

DNA-bait yeast strains were generated as previously described[60] and as follows (see Supplementary Table 1 within the Supplementary Data file). Promoters of 83 genes with a known association with cancer, incorporating ~2 kb upstream of the transcription start site, were amplified from human genomic DNA (Clonetech) using primers with Gateway tails (see Supplementary Table 1 within the Supplementary Data file). Promoters were first cloned into the pDONR-P4P1R vector using BP Clonase (ThermoFisher #11789100) to generate Gateway entry clones. Sequences were confirmed via Sanger sequencing. Each promoter was then cloned into the pMW#2 (Addgene #13349) and pMW#3 (Addgene #13350) destination vectors using LR Clonase (ThermoFisher #11791100), where they were inserted upstream of the *HIS3* and *lacZ* reporter genes, respectively. Destination vectors were linearized with single-cutter restriction enzymes (New England Biolabs R0520L, R0146L, R3127S, R0581S, R0193L, R0114S, R0187S, R0519L).

The pWM#2 and pWM#3 plasmids for each promoter were integrated simultaneously into the Y1Has2 yeast genome as previously described[18] and as follows. Yeast were inoculated in 1 L liquid YAPD media to a concentration of OD600 = 0.15 and were then incubated at 30 °C shaking at 200 rpm until they reached OD600 = 0.5, washed with sterile water, and washed again with 1X TE + 0.1 M lithium acetate (TE/LiAc). Yeast were resuspended in TE/LiAc with salmon sperm DNA (ThermoFisher 15632011) at a dilution of 1:10 before adding 2 μg digested plasmid. Six volumes of TE/LiAc + 40% polyethylene glycol were then added and samples were mixed gently ten times. Yeast were incubated at 30 °C without shaking for 30 min followed by 42 °C for 20 min, then resuspended in sterile water. Integrated yeast were plated on selective media lacking histidine and uracil to select for double integrants.

All clones and yeast strains are available upon request made to corresponding author J.I.F.B., and will be shipped within 1 month of request.

## Sequence confirmation of TF-prey and DNA-bait yeast strains

TF-pair prey and DNA-bait yeast strains were sequence-confirmed using the SWIM-seq protocol[36]. In brief, yeast were treated with zymolyase (0.2 KU/mL) (United States Biological Z1004) for 30 min at 37 °C followed by 10 min at 95 °C to disrupt cell walls and release DNA. TF ORFs and DNA-baits were PCR-amplified in 96-well format using forward primers with well-specific barcodes. For TF-prey, one set of primers was designed so that they targeted both the pAD2μ-*TRP1* and pGADT7-GW-*LEU2* vectors. See primer design below:

Forward primer (TF-prey):
5'−AGACGTGTGCTCTTCCGATCT[barcode]TAATACCACTACAAT GGATGATGT−3'
Reverse primer (TF-prey):
5'−GGAGACTTGACCAAACCTCTGGCG−3'
Forward primer (DNA-baits, pMW#2):
5'−AGACGTGTGCTCTTCCGATCT[barcode]GGCCGCCGACTAG TGATA−3'
Reverse primer (DNA-baits, pMW#2):
5'−GGGACCACCCTTTAAAGAGA−3'
Forward primer (DNA-baits, pMW#3):
5'−AGACGTGTGCTCTTCCGATCT[barcode]GCCAGTGTGCTGGA ATTCG−3'
Reverse primer (DNA-baits, pMW#3):
5'−ATCTGCCAGTTTGAGGGGAC−3'

PCR reactions were conducted using DreamTaq Polymerase (ThermoFisher EP0705) under the following conditions: 95 °C for 3 min; 35 cycles of: 95 °C for 30 s, 56 °C for 30 s, 72 °C for 4 min; final extension at 72 °C for 7 min.

Amplicons from each 96-well plate were pooled and purified using the PCR Purification Kit (ThermoFisher K310002). Each pooled sample was prepared as a single sequencing library by the Molecular Biology Core Facilities at the Dana-Farber Cancer Institute; DNA was sheared using an ultrasonicator (Covaris) prior to tagmentation. Libraries were sequenced using a NovaSeq with ~10 million reads (paired-end, 150 bp) per library. Sequencing data can be found at the NCBI Sequence Read Archive at accession number PRJNA1015222.

## Bioinformatics analysis of TF-prey sequencing data

The quality of FASTQ files were assessed using FastQC v.0.11 and MultiQC[61] software. Demultiplexing and trimming of adapters, barcodes and primer sequences were carried out using cutadapt 4.1[62] with the following parameters: -e 0.2 -pair-filter = both -O 10 for pAD2μ; and -e 0.2 -pair-filter = both -O 20 for pGADT7 vectors.

A FASTA file of the nucleotide sequences of expected TFs, including all possible isoforms, was generated using the package BIOMART[63] in R. First, we obtained the isoform IDs considering "ensembl" as dataset, 'ensembl_gene_id' as filter, and 'ensembl_trancript_id' as attributes. We then used the getSequence() function to obtain the coding sequence for each isoform. The resulting FASTA file was indexed using *bwa index*[64] and alignment was performed using *bwa mem* with default parameters. Samtools 1.10[65] was used to sort, index, and convert from sam to bam files using parameters by default.

To quantify the number of reads aligned to the expected sequence in each well, we developed an in-house R script primarily based on Rsamtools functions. We considered only those reads that mapped a TF sequence with a primary alignment score greater or equal to 90% of the trimmed read length, allowing for less than 5% of mismatches. We then determined the number of reads aligning to the expected sequence in each well, considering either the forward or reverse reads, and considered a correct match if the gene with the most aligned reads match the expected gene. Most wells had over 90% of reads aligned to the expected sequence. For a TF-pair to be considered "sequence-confirmed," we required both TFs to be confirmed in the TF1-TF2 yeast strain, for TF1 and the empty AD2u vector to be confirmed in the TF1-empty strain, and for TF2 and the empty pGADT7 vector to be confirmed in the TF2-empty strain. Additional positions in the arrays were verified by Sanger sequencing. Using these criteria, we

confirmed 297/508 TF-pair series for which yeast strains had been generated.

Code for this analysis can be found in the section "Bioinformatics analysis of TF-prey sequencing data" within https://github.com/jfuxman/PY1H_NatComm2023/.

## pY1H screening

Screening of TF-pairs and DNA-baits was performed similarly to eY1H screens as previously described[60] and as follows using a high-density array ROTOR robot (Singer Instruments). The five-plate TF-pair yeast array and DNA-baits were mated pairwise on permissive media agar plates and incubated at 30 °C for 1 day. Mated yeast were then transferred to selective media agar plates lacking uracil, leucine, and tryptophan to select for successfully mated yeast and incubated at 30 °C for 2 days. These selection plates were imaged and analyzed to identify array locations with failed yeast growth, which were then removed from further analysis. Diploid yeast were finally transferred to selective media agar plates lacking uracil, leucine, tryptophan, and histidine, with 5 mM 3AT and 320 mg/L X-gal. Readout plates were imaged 2, 3, 4, and 7 days after final plating. Yeast plate images are available at https://doi.org/10.7910/DVN/GITY2H[66].

## Image processing

To analyze the pY1H images we developed an open-source analyzer called DISHA (Detection of Interactions Software for High-throughput Analyses), in honor of Disha Patel who was very loved and passed away too soon. DISHA uses classical computer vision algorithms and deep-learning approaches to accelerate the analysis of pY1H readout plates. The overall pipeline of DISHA (Supplementary Fig. 2) includes, in this processing order, boundary cropping, grid generation, and colony segmentation algorithms. The boundary cropping algorithm converts the input image to grayscale and rescales the image intensity (blue color due to β-galactosidase activity) to enhance the yeast colonies from the background. Then an approximate binary mask of the colonies is created using a fixed threshold value. The plate boundary cropping is performed by limiting the region of interest to the first and last white pixel encountered vertically and horizontally in the binary mask. This is followed by the grid generation algorithm to localize the yeast colonies further and assign coordinates to each set of quadruplicate colonies based on a 1536 colony format (Supplementary Fig. 2). An approximate segmentation mask for the colonies is obtained through a sub-optimal subtraction of the plate background performed by a smoothing operation, followed by dynamic contrast stretching and convolving using edge detection kernels. The resulting mask is projected horizontally and vertically (Supplementary Fig. 2). The centers of the colonies are detected by zero-crossing analysis of the gradients of the projections (Supplementary Fig. 2). Given that equally spaced pins are used for yeast transfer, we assumed that the colonies are equidistant from each other, and therefore, we can extrapolate the grids based on the centers. A UNet-based segmentation model[67] was trained on our curated yeast segmentation dataset. Briefly, a fixed-size patch was randomly selected from pY1H assay images and generated multiple segmentation maps by varying the parameters of our manual segmentation pipeline. This dataset was curated by manually discarding the incorrect segmentation maps.

The size and intensity of the colony can be considered a proxy for reporter activity and used to determine cooperativity or antagonism between TFs. The area is computed by counting the number of non-zero pixels in a region identified as a colony. The intensity is computed by removing the background pixels from the region of interest and adding all the remaining pixel intensities. We further normalize this value by the area of the corresponding colony. Then a reporter signal score is calculated as follows (Eq. 1) that combines both area and intensity metrics of the TF pairs normalized by the average metrics from multiple empty-empty pairs (neither vector expresses a TF).

$$RS_{TF1-TF2} = [(I - I_{min}) \times A]_{TF1-TF2} - AVG([(I - I_{min}) \times A]_{empty-empty}) \quad (1)$$

Here, $I$ is the intensity, $I_{min}$ is the minimum non-zero intensity, and A is the area of the colony.

Using this reporter signal we generate three indices: Cooperativity index, Antagonism Index 1, and Antagonism Index 2. They are defined as follows (Eqs. 2–4).

$$Cooperativity\ Index = RS_{TF1-TF2} - RS_{TF1-empty} - RS_{empty-TF2} \quad (2)$$

$$Antagonism\ Index_1 = RS_{TF1-empty} - RS_{TF1-TF2} \quad (3)$$

$$Antagonism\ Index_2 = RS_{empty-TF2} - RS_{TF1-TF2} \quad (4)$$

DISHA also incorporates a visualization tool to represent the data generated by the analyzer more intuitively (Supplementary Fig. 3). This includes a Plate view that shows a segmented plate image where colonies can be selected and filtered by single-TF or TF-pair, and a Table view that displays a colony image comparison for each TF-pair with the corresponding single-TFs as well as area and intensity metrics.

Code and instructions for running the DISHA software can be found in the section "DISHA" within https://github.com/jfuxman/PY1H_NatComm2023/.

## Calling interactions

TF-pair strains were sorted based on each index (cooperativity, antagonism index 1, and antagonism index 2) separately. Images were then manually analyzed to call cooperative and antagonistic interactions. To call an interaction, we required the following criteria:

1. TF-pair, TF1, and TF2 yeast strains all showed growth in the mating selection plates prior to transfer to readout plates.
2. On readout plates, ≥3 out of 4 quadruplicate colonies were uniform for TF-pair, TF1, and TF2 yeast strains.
3. For cooperative interactions, TF-pair yeast showed a strong or moderate reporter activity relative to the empty-empty strain. TF1 and TF2 yeast showed only weak or very weak reporter activity.
4. For antagonistic interactions, TF1 and/or TF2 yeast showed a strong or moderate reporter activity relative to the empty-empty strain. TF-pair yeast showed only weak or very weak reporter activity.

See Supplementary Tables 4, 8, 13, and 16 within the Supplementary Data file for pY1H results.

## Literature overlap

Overlap of pY1H interactions with existing literature was determined using the CytReg2.0 database[26]. If CytReg2.0 reported at least one piece of evidence for binding of a TF to a cytokine promoter or regulation of the cytokine by the TF, then the TF-cytokine interaction was considered to be previously reported. To compare with eY1H data, we determined whether the TF had been found to bind the same cytokine promoter DNA-bait sequence tested in both eY1H and pY1H assays. Results from eY1H and pY1H assays were both compared to CytReg2.0 data after removing eY1H interactions already reported in CytReg2.0.

## Comparing eY1H and pY1H ChIP-seq overlap

The eY1H dataset consisted of 270 TF-promoter pairs, while the pY1H dataset contained 256 pairs derived from this study. We utilized the GTRD database to obtain ChIP-seq data for PDIs detected in the eY1H dataset (See "Overlap between ChIP-seq and pY1H interactions" for more details). Subsequently, we excluded TF-promoter pairs for which ChIP-seq information was not available. To compare the proportion of

TF-promoter pairs with ChIP evidence between eY1H and pY1H, we employed a two-tailed proportion comparison test and calculated a standard error of proportion using the following equation (Eq. 5) where $p$ = proportion and $n$ = sample size:

$$SE = \sqrt{p(1-p)/n} \qquad (5)$$

We also performed a network randomization analysis separately for eY1H and pY1H datasets. For each dataset, we generated 10,000 networks and performed 20,000 edge-switches to assess the significance of the observed results (See: "Network randomization analysis"). Based on the 10,000 random networks generated, a Z distribution was used to obtain a Z-scores and two-tailed $p$-values for the original eY1H and pY1H networks.

Code for obtaining the ChIP-seq data can be found in the section "Obtaining ChIP-seq data from GTRD" within https://github.com/jfuxman/PY1H_NatComm2023/. Code for the randomization analysis can be found in the section "Network Randomization Analysis (eY1H and pY1H with ChIP data)" within https://github.com/jfuxman/PY1H_NatComm2023/.

### Overlap between ChIP-seq and pY1H interactions

The ChIP-seq peaks mapping to the cytokine promoter sequences tested by pY1H assays were obtained from GTRD database[68] considering the following filters: peaks calling = MACS2, reference genome = hg38, format file = bigBeds. A TF was considered to be binding a cytokine promoter if the summit point of any significant peak ($p - value \leq 10^{-4}$) was located within the promoter's genomic coordinates. The output was a table showing the peak of the TF, its genomic coordinates, and the cell line used. TF-pairs detected by pY1H assays for which ChIP-seq data was available for both TFs were further considered. For each TF-pair interaction with a cytokine promoter, evidence for co-binding was considered when both TFs had ChIP-seq peaks within the corresponding promoter, either in different or the same cell line, and the peak summits were within 50 bp of each other.

Code for this analysis can be found in the section "Obtaining ChIP-seq data from GTRD" within https://github.com/jfuxman/PY1H_NatComm2023/.

### Identification of binding sites of TF-pairs in cytokine promoters

Position Weight matrix (PWM) motifs were downloaded from CISBP 2.0 database[69] for each TF. PWM motifs with all sites probabilities <0.8 were removed to reduce low-specific motifs. To determine if a PWM motif was present within a promoter sequence, we calculated the sum of log odds for each position in each promoter using the following formula (Eq. 6):

$$Score(s, PWM) = \sum_{t=0}^{|s|-k} \prod_{i=1}^{k} \left( \frac{PWM_i[s_{t+i}]}{p_i} \right) \qquad (6)$$

Where $i$ = 1,2,3,4 corresponding to {A,T,C,G}, $p_i$ is the background frequency of such nucleotide, which is 0.25. $k$ = length of the PWM, $|s|$ = length of the sequence. Each score was converted to a $p$-value using the TFMsc2pv function from the TFMPvalue package[70]. Motifs were filtered considering a $p - value \leq 10^{-4}$. As many motifs for the same TF were very similar, we merged all motifs for a TF that overlapped with each other using the following steps:

1. Consecutive motifs for a TF within a DNA-bait sequence that shared 80% or more nucleotides were labeled into the same group.
2. For each group of overlapping 'n' motifs within a DNA-bait, we selected the sub-region corresponding to the intersection

between all n motifs, only if this sub-region was four nucleotides or longer and named this as 'core motif'.
3. If the intersection region was shorter than four nucleotides, we repeated the process by taking the intersection region shared by 'n-1' motifs.

This algorithm produces a set of non-overlapping core motifs of a TF within DNA-bait sequences. We manually reviewed the final list of core motifs to ensure that it was unique and did not overlap with others. To compare with pY1H interactions, a TF-pair was considered to potentially binding a DNA-bait if a core motif for each single-TF was present in the DNA-bait within 10 nt of each other.

Code for this analysis can be found in the section "Identification of binding sites of TF-pairs in cytokine promoters" within https://github.com/jfuxman/PY1H_NatComm2023/.

### Network randomization analysis

The significance of overlap between TF-pairs determined by pY1H assays and those presenting ChIP-seq peaks within the same promoter was evaluated by a network randomization analysis. First, we built a directed network graph where the source node was ($TF_1-TF_2$), and the target node was cytokine promoter used in the pY1H screen. Then, 10,000 networks were generated by performing 20,000 edges-switches while maintaining the same degree for each node[71] using the igraph package in R.

For the original pY1H network and each of the randomized networks, we determined the number of edges overlapping with the ChIP-seq data. Based on the 10,000 random networks generated, a Z distribution was used to obtain Z-scores and two-tailed $p$-values for the original pY1H network. This analysis was performed considering: (1) ChIP-seq peaks found in the same cell line, and (2) ChIP-seq peaks found in different cell lines.

A similar randomization analysis was performed to compare pY1H interactions with TF motifs found in the corresponding cytokine promoters. We evaluated the significance of detecting binding sites for both TFs anywhere in the promoters and within 10 bp from each other.

Code for ChIP-seq overlap randomization analyses can be found in the section "Network Randomization Analysis (ChIP peaks)" within https://github.com/jfuxman/PY1H_NatComm2023/.

Code for DNA binding motif randomization analyses can be found in the section "Network Randomization Analysis (Promoters)" within https://github.com/jfuxman/PY1H_NatComm2023/.

### Data visualization and statistical analyses

Network visualizations were constructed using Cytoscape Version 3.9.1. Scatter plots, violin plots, histograms, bar graphs, and heat maps were generated using GraphPad Prism Version 9.

### Paralog partner similarity

TFs were classified based on their DBD family, as reported in Lambert et al.[58]. A pairwise alignment was performed using the BLOSUM62 matrix from the package seqinr, and the amino acid identity score was assigned to each pair of TFs from the same TF family. To determine if TFs with greater amino acid identity have similar functional relationships (antagonism and cooperativity) with their shared TF interactors tested by pY1H, we calculated the Jaccard similarity index as follows:

1. For a pair of TFs ($TF_a$, $TF_b$), we obtained the list of TF partners that were both tested by pY1H assays.
2. For each $TF_a$, we generated a binary vector (P$_{1,c}$, P$_{1,a}$, P$_{2,c}$, P$_{2,a}$,...), where P$_{i,c}$ indicates whether partner i has at least one cooperative interaction involving $TF_a$, (true = 1, false = 0), and where P$_{i,a}$ indicates whether partner i has at least one antagonistic interaction involving $TF_a$.

3. Then the Jaccard index was determined as the number positions with 1 in both $TF_a$ and $TF_b$ vectors divided by the number of positions with a 1 in either $TF_a$ and $TF_b$ vectors.

The Jaccard score ranged from 0 to 1, where 1 indicate both TFs ($TF_a$, $TF_b$) have the same functional relationships with the same partners and 0 indicates both TFs have completely different functional relationships with their shared partners.

The percent amino acid identity was classified in three groups: Low identity (<30%), Medium identity (30–50%) and high identity (>50%). A Mann–Whitney's $U$-test was performed to evaluate significant differences between groups regarding paralog partner similarity based on the Jaccard index.

Code for determining similarity of interaction patterns between paralogs can be found in the section "Paralog partner similarity" within https://github.com/jfuxman/PY1H_NatComm2023/.

### TF expression analysis

The single cell RNA-Seq data was obtained from the Tabula Sapiens atlas[37] (see Supplementary Table 9 within the Supplementary Data file). To avoid technical confounding factors, only samples that were generated by 10X Genomics protocols were used. After obtaining the data, cells with no less than 500 genes, no more than 7500 genes, no more than 10,000 UMIs, and no more than 25% mitochondrial contents were kept for the downstream analyses. The normalized counts per cell were generated by dividing the gene counts per cell by the total number of UMIs per cell and then multiplied by 1,000,000, to determine the counts per million (cpm). After log normalizing the cpms and conducting a principal component analysis, Harmony[72] was used to remove batch effects. Then, the k-nearest neighbor graph was constructed between cells and the Louvain community clustering was used to cluster cells based on the constructed graph. A total of 187 clusters across samples were identified. All the steps above were performed by Seurat in R environment[73]. Differential expression analyses (Wilcoxon ranked sum test) were performed between clusters to identify the genes that were significantly upregulated in each cluster. The genes with false discovery rates <0.05 were used to compare with the gene markers curated in the CellTypist[74] database to assign cell types to clusters in each sample.

Code for assessing TF expression from the Tabula Sapiens atlas can be found in the section "TF expression analysis" within https://github.com/jfuxman/PY1H_NatComm2023/.

### Tissue/cell type expression specificity scoring of genes

To study the gene expression specificity among cell types and tissues, a tissue/cell type expression specificity score (TCESS) was calculated for each TF adapting a previously entropy-based approach to single-cell RNA-seq data[75] (see Supplementary Table 10 within the Supplementary Data file). Briefly, given a cluster C, which had n cells, the total expression of $TF_a$ was calculated using the following formula (Eq. 7):

$$\text{Exp}_{\text{TF}_a}^C = \left( \sum_{\text{Cell} \in C}^{\text{Gene} = \text{TF}_a} \exp_{\text{Gene}}^{\text{Cell}} \right) + 1 \tag{7}$$

Then the TCESS was calculated as follows (Eq. 8):

$$\text{TCESS} = \sum_{\text{TF}_a}^{C \in \text{dataset}} \left( \frac{\text{Exp}_{\text{TF}_a}^C}{\text{sum}\left(\text{Exp}_{\text{TF}_a}^C\right)} \right) * \log_2 \left( \frac{\text{Exp}_{\text{TF}_a}^C / \text{sum}\left(\text{Exp}_{\text{TF}_a}^C\right)}{\text{mean}\left(\text{Exp}_{\text{TF}_a}^C / \text{sum}\left(\text{Exp}_{\text{TF}_a}^C\right)\right)} \right) \tag{8}$$

The TCESS ranges from 0 when $TF_a$ expression is identical across all clusters to log2(#clusters), in this case ~7.54, when $TF_a$ is expressed exclusively in one cluster.

Code for calculating TCESS scores can be found in the section "TF expression analysis" within https://github.com/jfuxman/PY1H_NatComm2023/.

### Transcription factors co-expression among tissue/cell types

To study the co-expression patterns of pairs of TFs across cell types/tissues, a scoring system based on the Simpson Index was developed[76]. In a given cell type/tissue cluster, if the cpms of a given TF in the cluster was >10% of the maximum cpms for the TF across all clusters, the TF was considered 'expressed' in the given cluster. For example, if the $TF_a$ in a cluster B is 1.2 cpms, and the maximum expression of $TF_a$ across all clusters is 10 cpms, then $TF_a$ is considered to be expressed in cluster B. Then, for each $TF_a$, we generated a binary vector indicating whether $TF_a$ was expressed in each of the 187 cell clusters. Finally, for every pair of TFs we determined the co-expression score using the Simpson index, by dividing the number of clusters expressing both TFs by the number of cluster where the most tissue specific TF is expressed.

Code for determining co-expression of TF-pairs can be found in the section "TF expression analysis" within https://github.com/jfuxman/PY1H_NatComm2023/.

### Structural predictions of STAT1/STAT3 dimers

We utilized AlphaFold 2 to generate the structures of STAT3-203, STAT1-201, and STAT1-202, employing the following parameters: --model_preset = monomer and --db_preset=full_dbs. To visualize the structures, we utilized Pymol and selected the surface and cartoon representations. Parallel and antiparallel conformations of dimers were arranged manually in Pymol.

### Statistics and reproducibility

No statistical method was used to predetermine sample size. The number of DNA regions of interest selected for screening was based on feasibility considerations gleaned from previous experiments. When generating protein-pair arrays, we started with all protein-pairs known or suspected to interact with one another, and we report data corresponding to all pairs for which yeast strains were sequence-confirmed.

As established prior to data collection, data were excluded for a protein-pair if the protein-pair or either corresponding single-protein yeast strain were deemed "inconclusive" by one or more of the following criteria: yeast strain was not sequence-verified; yeast strain did not show adequate growth in the array; yeast strain did not display at least 3 uniform colonies; yeast strain was contaminated during screening.

As demonstrated in our Supplementary Information, we conducted two replicate screens for the CCL15 promoter and observed a high level of reproducibility in event calling. Replicate screens of other promoters were also successful. All interactions are tested in quadruplicate colonies and we require uniform reporter signal from 3 out of 4 replicate colonies for an interaction to be considered.

We randomized locations of yeast strains in our array plates so that strains expressing a given protein were dispersed throughout the plate. Similarly, we distributed "empty" control yeast strains, which were used for normalization, throughout each array plate to avoid any biases that might arise based on plate location.

As no group allocations were involved in this study, researcher blinding was not applicable. Although researchers were not blinded to the identity of yeast strains during analysis, unbiased results were ensured as follows. First, yeast strains were sorted according to objective cooperativity and antagonism indexes prior to manual curation to generate an initial event list. Second, researchers used an unlabeled full plate layout view to blindly identify "positive" colonies. The initial list of cooperative and antagonistic events was then further curated to include only those that involved blindly selected "positive" colonies.

## Reporting summary

Further information on research design is available in the Nature Portfolio Reporting Summary linked to this article.

## Data availability

All data generated during this study are included in this published article and its Supplementary Information/Source Data files. Sequencing data can be found at the NCBI Sequence Read Archive at accession number PRJNA1015222. Yeast plate images are available at https://doi.org/10.7910/DVN/GITY2H[66]. Enhanced yeast one-hybrid data can be found at https://doi.org/10.1093/nar/gkaa1055[26]. The CytReg database can be found at https://cytreg.bu.edu/search_v2.html. ChIP-seq data were obtained from the GTRD database (https://doi.org/10.1093/nar/gkaa1057; http://gtrd.biouml.org/)[68]. DNA binding motif data were obtained from the CIS-BP database (https://doi.org/10.1016/j.cell.2014.08.009; http://cisbp.ccbr.utoronto.ca/)[69]. Expression data were obtained from the Tabula Sapiens atlas (https://doi.org/10.1126/science.abl4896)[37]. All clones and yeast strains generated in this study are available upon request made to corresponding author J.I.F.B., and will be shipped within 1 month of request. Source data are provided with this paper.

## Code availability

All custom code used to generate and analyze data in this study is available at https://github.com/jfuxman/PY1H_NatComm2023 and https://doi.org/10.5281/zenodo.8329035[77].

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

## Acknowledgements

This work was funded by the National Institutes of Health grants R35 GM128625 awarded to J.I.F.B and U01 CA232161 awarded to J.I.F.B and M.V. We thank Dr. Trevor Siggers for critically reading and commenting on the manuscript.

## Author contributions

A.B. and J.I.F.B. conceived the project. A.B. and R.L. performed the pY1H screens with contributions from Y.C., S.S., C.S., X.L.. A.B., J.I.F.B., L.F.S.-U., and Z.L. performed data analyses. M.P. and C.C. developed DISHA. K.S., T.H., M.V., and D.E.H. provided human TF and isoform ORFs. A.B. and J.I.F.B. wrote the manuscript with contributions from L.F.S.-U., Z.L., M.P., and C.C. All authors read and approved the manuscript.

## Competing interests

The authors declare no competing interests.
