## [Peer Review File · Nature Communications]

REVIEWERS' COMMENTS

Reviewer #1 (Remarks to the Author)

The authors present an expanded version of the eY1H assay to be able to test for heterodimer interactions with target DNA sequences. This new experimental approach addresses the common concern with Y1H studies that it misses interactions beyond TFs that function as singletons or homodimers. Using a data driven approach, they narrow down TFs that have been demonstrated to or are likely to participate in protein-protein interactions to influence gene expression of targets. The authors clearly present their motivation for developing this assay and show that it works well - recapitulating previously published data and also finding novel interactions with tested heterodimers combinations. They categorize these interactions as cooperative, antagonistic or complex. In excellent experimental design, they use two formats - AD1 and AD2 to determine the type of antagonistic interactions. They develop an image analysis tool to quantify the reporter readouts to better assign strength of the readout for comparison purposes. It was interesting to see how viral transcriptional regulator proteins can alter TF binding. This was a creative way to show that this assay can be used for a wider variety of studies.

Concerns

One major question I have is if reporter read out strength in Y1H assays has been correlated with binding strength in vivo? The authors hinge a lot of interpretation of the cooperative binding readout to stronger reporter response on this assumption but I am unaware of any studies that have thoroughly investigated this correlation.

Why were 200 genes lost in the yeast validation process? This is a significant amount.

Line 37: What is indirect cooperativity? There is no citation and no mention of this elsewhere in the paper.

Line 148-149: were all the genes/DNA regions in these "previously undetected interactions" screened in eY1H assays?

Line 194-198: What are the numbers for this data? The percentage and total interactions show both be presented.

321- 322: For how many total interactions?

Minor comments:

Line 34: replace with 'some TFs can positively or negatively...'

Line 156: typo in the word 'how'

Figure comments:

Many of the figures are dark, overly crowded and hard to read. The colors for the nodes should be lighter (dark purple is too dark) and using a different font will make the text clearer.

Figure 1:

Error in figure 1c in the bottom center column (path should not be there)

1d needs to be 3 panels

1e - why is this presented in percentage instead of total count? What interactions is this summarizing? Plotting this as a bar chart (and not stacked) will make this information more clear.

Figure 2:

2e (and associated text) - is it expected for there to be so little overlap between the eY1H and pY1H?

2f is very unclear to me. What are the error bars? Is this test biased due to the nature of selection of genes you've screened for pY1H - intensively studied pathways/genes that have known interactors? Are the number of interactions and targets for eY1H comparable? If the eY1H data

encompasses genes from less studied pathways or a much larger number of interactions, this chart is a misrepresentation of how well these assays can be validated in vivo. Removing this chart and in the text stating the percentage (and total number) of pY1H PDIs found with literature support is the more straightforward way to present this data.

Figure 3

3e - is there a way to summarize this data in a more interpretable manner? Maybe a density plot? That data is too overlapping for this to be clearly read.

Figure 4

4a - the light green/white text combo is difficult to read.

4b - this panel is too much to read and not labeled sufficiently. I recommend using 1 or 2 of these plots, moving the rest to supplement, and labeling much more. Not knowing the baits removes a lot of important information from this figure. Also, the arrows are almost impossible to see.

Reviewer #2 (Remarks to the Author)

Revision of NCOMMS-23-18312-T

The manuscript by Berenson et al. introduces paired yeast-one hybrid (pY1H), an experimental technique to screen transcription factor (TF) interactions with DNA in a high-throughput manner and assess potential mechanistic processes linked to biomolecular interactions. This assay is of major relevance for the Systems Biology / Functional Genomics community, given the need of automating the screening of TF pairs in a controlled system and allowing the genetic permutation of hundreds of TFs and target sequences.

To the best of my knowledge (Computational Biology and Biochemistry), this setup is a relevant, major improvement to the previous eY1H approaches previously developed by some of the same authors, and allows screening and mechanistic assessment of biological interactions between TF pairs e.g. competition/cooperativity/sequestering TFs. Overall, the main sections of this manuscript are very well written and presented, which makes the main work easy to read, understandable, and relevant to a broad audience.

I am listing below revisions, mostly related to data analyses, limitations, and data/code release, which in my opinion would strengthen the interpretation of this work and the accessibility of experimental data generated. I am supportive of the publication of this work, once these points are assessed for revision and reviewed,

Major revisions:

1.

(i) The distinction from observed results "competition" and "sequestration" interactions of TF1 by TF2, among others, does not seem to be the only theoretically plausible in some cases. If TF2 is not by itself able to activate the reporters, a "sequestration" annotation could be given to TF1+TF2, despite perhaps being in reality and actual a competition event, in which where TF2 by itself is faulty and unable to activate HIS3/LacZ. (ii) As an additional observation, it looks like both 1-AD and 2-AD are always required to mechanistically interpret the observed signals.

2.

These two ideas, and potentially other mechanistic limitations the authors can identify in other cases, are necessary to be further discussed in the Discussion. Some panels might require updates to contain both 1-AD and 2-AD designs, in case they are required for interpretability (e.g. 4c, S1). Fig 2g: (i) The analysis of interactions with ChIP-seq seems descriptive, and a statistical test could be required. (ii) In the previous panel (2f) a comparison/enrichment comparison with eY1H was performed. Can the authors provide such a comparison and interpretation of whether the agreements of pY1H with ChIP-seq are higher than eY1H? As an additional dataset, authors could also utilize TF pairs reported by Jolma et al. (CAP-SELEX, Nature 2015), to study whether interfamily TF1-TF2-DNA complexes are also more strongly supported than eY1H.

3.

It is my impression that pY1H datasets, and the complementary eY1H datasets, are very interesting for the ML/Genomics community, for the purpose of predicting signals based on TF

sequence, DNA sequence, and interactions with other TFs, using Deep Learning models e.g. Graph-Neural networks with sequence representations. The current Supplementary Table format is generic for publication, yet it seems that a processed dataset with normalized signals per replicate, plus harmonized annotations, could allow other researchers to use this data and try to model observations using mechanistic and/or generative approaches.

4.

Line 154: "Overall, this screen-detected novel instances of sequence-specific cooperativity and antagonism...". I think the interpretation of sequence-specific and/or TF-specificity effects in this work is limited. I would either replace "sequence-specific" statements with "gene interactions" whenever suitable. Alternatively, I think it could be better to exemplify how these interactions are suggested to happen at the DNA sequence level e.g. are promoters with strong/weak bZIP motifs still preferred for cooperative interactions? Are cooperative/antagonistic promoters showing a higher/lower number of motifs for certain TF families? Results could also be interpreted by showing mapped motifs and specific sequence modifications. Examples for either 1d, 2d, 4d (XCL1 and TNF5F8), or 5c (any hTF-vTR) could easily generate and enhance the interpretation of results. Authors could decide on at least two examples from that list, or other suitable ones, for interpretation.

5.

LN308: "STAT3 and STAT1 affect the equilibrium between STAT3/STAT3 homodimers" Here it would be nice to clarify if the authors interpret that these complex events are happening once either STAT3 or STAT1 are binding to DNA, or in a DNA-independent manner. A protein structure visualization of those cases, highlighting activation domains and/or deleted regions, or the general scheme of this case as a supplementary figure, could be useful for the interpretation of this point.

6.

The provided code repository (LN568 <https://github.com/mahir1010/D.I.S.H.A> and DISHA viewer) is weak. Methods sections require rewriting and linking to the right processing notebook/scripts, with an example dataset e.g. "Image processing" As currently presented, the code does not allow reproducing the results from this work. I please request the authors to present simplified documentation notebooks (e.g. Jupyter notebooks) showcasing the most relevant data analyses conducted in this work (summary barplots, ChIP-seq, network permutations, TCESS comparisons), with a processed dataset that is retrievable, and with minimal installation dependencies. This does not require a software release and can be mounted into a reproducibility-specific GitHub repository, per figure and/or panel.

Minor.

LN 39: This claim makes sense, but a reference to a TF-paralog competition example is needed.

LN 46: I think a reference to the futility theorem review from Wasserman's lab is also needed here (PMID 15131651). The current reference seems to be an analysis based on that tool

LN 46: "Predictions are generally more challenging for heterodimers" This seems to be mainly due to wet lab, experimental limitations. To my knowledge, there's no benchmark on whether computational predictions are more challenging for hetero- than heterodimers motifs. Provide reference, If any.

LN 75: "Further expanding the TF landscape", I think "TF interactome landscape" might be a suitable term there, unless the authors are referring to something else.

LN147: Shortly clarify in text/captions hat how these literature interactions are gathered related to cytokine-TF. I understand it's a single paper by Santoso et al., once getting into the references.

LN179: caption/text needs to indicate if the statistical test is one- or two-tailed.

3C: "other pairs" instead of TF pairs.

3D: The number of observations in each violin plot would be relevant to know.

2d: The individual number of coop. and antagonistic interactions are not clear in this barplot. I recommend visually separating those two, and highlighting percentages as well e.g. all bars higher or equal to one = 63%.

Figure 5b: It is unclear if the heatmap enumerated all interactions tested and or found. Can the authors describe in results/captions why not all vTR and hTFs interactions were tested e.g. were STAT vs. vTFs interactions tested, or no interactions were found? Highlighting with labels non-zero heatmap cells the light cells could be suitable to highlight rare cases with only "1" or "2" observations.

Reviewer #3 (Remarks to the Author)

Comments for Authors:

In the manuscript entitled "Paired yeast one hybrid assays to detect DNA-binding cooperativity and antagonism across transcription factors," the authors present not only a novel method to investigate the interplay of transcription factors (TFs) in gene regulation but also a multitude of data obtained by different settings analyzed in the study that can be used by other researchers as starting point for future studies. The abstract and the main body of the manuscript are written of overall clarity and readability and the figures are very well-designed. The method allows the investigation of transcriptional regulation of a specific promoter sequence by a defined pair of transcription factors. The findings suggesting that the role of a TF depends on its TF partner as well as on the target DNA sequence. Although no in-depth analysis was performed for a specific TF pair (e.g. TF interaction/degradation studies, in mammalian cells), the method and the results presented are of immediate interest to many people in the field of transcriptional regulation including cell biologists, oncologists, immunologists, and virologists investigating virus host interactions on the level of cellular and viral gene expression that influences virus replication and virus induced oncogenesis. Therefore, this study represents a major advance in a broad scientific field.

Key results: The study provides a novel method that is broadly applicable and data that shed light into TF cooperativity and antagonism, isoform usage and explains the importance of the cell type specific background for the understanding of the mode of action of a specific TF on a defined DNA sequence. This study addresses a critical aspect of transcriptional regulation by focusing on the cooperation and antagonism between TFs. The topic is highly relevant to a broad readership, as it contributes to our understanding of the complex mechanisms governing gene expression. The authors' novel approach represents a significant advancement in the field and opens up new avenues for studying TF interactions.

Validity of the approach: The manuscript provides a comprehensive analysis of the proposed method's efficacy by testing it on the well-established examples, AP1 and NFkB subunits, on cytokine promoter regions. This validation step is crucial to demonstrate the robustness and reliability of the pY1H assays. Furthermore, the authors perform a broad screening experiment including almost 300 TF pairs, contributing to our understanding of cooperativity and antagonism across specific transcription factors which can explain to some extent the complexity of tissue-specific gene regulation. This aspect highlights the versatility and potential applications of their method in uncovering the interplays of TF interactions across various biological contexts.

An additional strength of this manuscript lies in the authors' comparison of the effects of TF isoforms and viral factors on cellular TFs. These comparisons offer valuable insights into the functional consequences of TF variations and viral-host interactions on specific promoter sequences. By exploring these diverse scenarios, the authors provide a comprehensive evaluation of their method's capabilities and showcase its broad utility. In addition, limitations of the methods are also clearly stated by the authors. The data interpretation is robust, valid and reliable. The conclusions drawn by the authors are comprehensible.

After a careful evaluation of the manuscript, I am pleased to recommend it for publication with only minor changes.

To improve the manuscript, I suggest addressing the following minor points:

Interestingly, pY1H cooperative events significantly overlapped with motif predictions and ChIP-seq data while antagonistic TF-pairs are less predictable. The authors explain this with sequestration rather than competitive binding of both TFs.

Here, the authors should also mention that viral factors such as HPV16 E7 are potent degraders of cellular TFs such as pRB, MYOP or PTPN14. In their assay the authors could determine the putative degradation of TFs in yeast co-expressing viral proteins and protein quantification. Alternatively, consider expanding the discussion section to elaborate on this scenario.

Specify/label viral proteins in Figure 5 (e.g. "HPV16 E7" instead of "HPV16"). This will facilitate readers' comprehension of the figure.

I also recommend naming the virus families in the abstract (HPV, EBV, AdV, HIV, HTLV, HBV) in order to strengthen the attention of the findings by virologists.

Line 156: "how"

In summary, the manuscript is of high interest to a broad readership. The authors' significant contributions to the understanding of transcriptional regulation, including the exploration of TF isoforms and viral factors, make this study highly impactful. With the suggested minor revisions, this manuscript will be an excellent addition to Nature Communications, advancing the field of gene regulation and inspiring further research in this area.

Luise Florin, Mainz, Germany

RESPONSE TO REVIEWERS' COMMENTS

Reviewer #1

The authors present an expanded version of the eY1H assay to be able to test for heterodimer interactions with target DNA sequences. This new experimental approach addresses the common concern with Y1H studies that it misses interactions beyond TFs that function as singletons or homodimers. Using a data driven approach, they narrow down TFs that have been demonstrated to or are likely to participate in protein-protein interactions to influence gene expression of targets. The authors clearly present their motivation for developing this assay and show that it works well - recapitulating previously published data and also finding novel interactions with tested heterodimers combinations. They categorize these interactions as cooperative, antagonistic or complex. In excellent experimental design, they use two formats - AD1 and AD2 to determine the type of antagonistic interactions. They develop an image analysis tool to quantify the reporter readouts to better assign strength of the readout for comparison purposes. It was interesting to see how viral transcriptional regulator proteins can alter TF binding. This was a creative way to show that this assay can be used for a wider variety of studies.

We thank the reviewer for the positive and encouraging comments and for the suggestions to improve figure and text clarity.

Concerns

One major question I have is if reporter read out strength in Y1H assays has been correlated with binding strength in vivo? The authors hinge a lot of interpretation of the cooperative binding readout to stronger reporter response on this assumption but I am unaware of any studies that have thoroughly investigated this correlation.

The reviewer raises an important point about the correlation between Y1H reporter strength and strength of in vivo binding. Of note, most of our claims relate to the relative binding strengths of monomers/homodimers and heterodimers to the DNA bait in yeast, which may not necessarily reflect binding in the endogenous genomic context in different cell types. Nevertheless, we have previously observed that eY1H signal intensities of TF variants correlate with luciferase reporter activity levels when validated in mammalian cells (PMID: 25910213, Figure 5b-c). This supports that, for a given TF, drastic differences in Y1H signal are likely to correspond to differences in binding strength. Furthermore, we were conservative in calling cooperative and antagonistic events. For cooperative events, we required either: A) obligate cooperative binding, where neither single-TF strain shows any signal and the TF-pair strain has a clear signal, or B) a drastic increase in reporter activity, e.g. from a very weak signal to a strong signal. For antagonistic events, we required either: A) complete antagonism, where a single-TF strain shows a clear signal and the TF-pair strain has no signal, or B) a drastic decrease in reporter activity, e.g. from a strong signal to a very weak signal. Obligate cooperative binding and complete antagonism account for >90% of pY1H events, minimizing reliance on signal strength comparisons. To make this point more clear, we have updated Supplementary Data 8

to include the colony area and intensities, so that readers can compare the strength of activity between monomer and pair strains.

We have added the following text in the Results section to clarify this for the reader (lines 99-104):

“This system of event calling is supported by two main findings. First, it was previously observed that eY1H reporter signal strength correlates with signal from more quantifiable binding reporter assays in mammalian cells”. Second, >90% of events detected in initial pY1H assays corresponded to obligate cooperative binding (where neither TF has any reporter signal in the absence of its partner) or complete antagonism (where a single-TF signal is completely lost in the TF-pair strain), minimizing reliance on signal strength comparisons.”

Why were 200 genes lost in the yeast validation process? This is a significant amount.

The number of TF-pairs lost during the validation process is the result of our stringent requirements for considering a TF-pair “sequence-confirmed.” We used a next-generation sequencing approach to confirm both TF clones for each yeast strain. To consider a TF clone “confirmed,” we required that the human gene with the greatest number of aligned reads matched the expected gene. For each TF1-TF2 pair, we required sequence confirmation for both TF clones in each of three yeast strains: the TF1-TF2 pair strain, the TF1-empty strain, and the TF2-empty strain; as well as confirmation of the empty vector in the TF1-empty and the TF2-empty strains. Therefore, for a TF-pair to be sequence confirmed, six individual clones needed to be confirmed. Using this approach for single-TF yeast strains has previously yielded a validation rate of ~85-95%. Therefore, when requiring confirmation for six clones, we would expect a validation rate of ~40-70%. We validated 58% of TF-pairs using this approach, which we found to be acceptable as it lies within the higher end of the expected range.

We have added the following text in the Methods section “Bioinformatics analysis of TF-prey sequencing data” (lines 486-491):

“For a TF-pair to be considered “sequence-confirmed,” we required both TFs to be confirmed in the TF1-TF2 yeast strain, for TF1 and the empty AD2u vector to be confirmed in the TF1-empty strain, and for TF2 and the empty pGADT7 vector to be confirmed in the TF2-empty strain. Additional positions in the arrays were verified by Sanger sequencing. Using these criteria, we confirmed 297/508 TF-pair series for which yeast strains had been generated.”

Line 37: What is indirect cooperativity? There is no citation and no mention of this elsewhere in the paper.

We have added a citation to PMID: 28349863 (Morgunova and Taipale, “Structural perspective of cooperative transcription factor binding,” Current Opinion in Structural Biology, 2017). This paper discusses the various mechanisms of cooperative TF binding, including those that do not involve protein-protein interactions, which we have termed “indirect cooperativity.” We have clarified this in the Introduction (lines 36-38): *“Some TFs bind DNA cooperatively, either via mutual cooperativity (e.g., as heterodimers or by indirect cooperativity mediated by DNA²), or when a DNA-bound TF recruits a second TF.”*

Line 148-149: were all the genes/DNA regions in these “previously undetected interactions” screened in eY1H assays?

From our pY1H cooperative binding events, we derived 80 PDIs between individual TFs and cytokine promoters. Of these, 71/80 had been tested by eY1H, none of which produced a positive binding signal, likely because the TFs tested function mostly as heterodimers. We have added the following underlined text to the Results text to clarify (lines 140-143):

“This suggests that pY1H can recapitulate known PDIs while revealing previously undetected interactions that require cooperativity, including 71 individual PDIs that were tested previously by eY1H and had shown no binding signal.”

Line 194-198: What are the numbers for this data? The percentage and total interactions show both be presented.

For cooperative TF events with available data, 55/137 (40%) had ChIP-seq peaks for both TFs in any cell line, and 25/106 (24%) had ChIP-seq peaks for both TFs in the same cell line. We have added the interaction numbers alongside the percentages in the results text as follows (underlined, lines 185-190):

“For 40% (55/137) of cooperative interactions with available data, both TFs have ChIP-seq peaks in the promoter in at least one cell line, a significantly greater overlap than expected for a randomized network. Furthermore, for cell lines with ChIP-seq data for both TFs, 24% (25/106) of cooperative interactions had ChIP-seq peaks for both TFs in the same cell line, which was also greater than expected for a randomized network.”

In addition, these numbers are now included in the pie charts in Figures 2h-i.

321- 322: For how many total interactions?

From our pY1H assay involving human TFs and viral proteins, we observed 8 cooperative events and 42 antagonistic events. We have added the following underlined text in the Results section (lines 291-292) to clarify: “*We observed both cooperativity (8 events) and antagonism (42 events) between 11 vTRs and 11 human TFs.*”

Minor comments:

Line 34: replace with ‘some TFs can positively or negatively...’

In line 34, we have changed “...TFs can positively or negatively affect one another’s ability to bind DNA” to say, “...some TFs can positively or negatively affect one another’s ability to bind DNA.” This clarifies that only a subset of human TFs are known to positively or negatively affect one another.

Line 156: typo in the word ‘how’

We have fixed this typo.

Figure comments:

Many of the figures are dark, overly crowded and hard to read. The colors for the nodes should be lighter (dark purple is too dark) and using a different font will make the text clearer.

We thank the reviewer for the suggestions to improve figure readability. We have made the following changes to make certain figure panels easier to read:

1. Figure 4a: We have changed the colors to lighter purple and green and changed all text to black for this panel. We also enlarged the nodes to make the labels easier to read.
2. Figure 3e: We have changed Figure 3e to make the data points more distinguishable by making the points small and uniform in size and displaying the expression similarity scores in the new Figure 3f.
3. Figure 4b: We have enlarged the figure to make it more easily readable and added DNA bait names.

Figure 1:

Error in figure 1c in the bottom center column (path should not be there)

We have fixed this screenshot error.

1d needs to be 3 panels

We have separated Figure 1d into Figure 1d (full NF- κ B/AP-1-cytokine network), 1e (NF- κ B subnetwork showing TF-TF relationships), and 1f (AP-1 subnetwork showing TF-TF relationships). Previous Figure 1e has been renumbered to Figure 1g. We have added the following underlined mentions of the new Figure 1e and Figure 1f to the results text (lines 145-148): “*This includes antagonism of REL by RELB at 4 cytokine promoters (**Fig. 1e**), consistent with findings that RELB/RELB and REL/RELB dimers display reduced DNA binding compared to other NF- κ B dimers^{28,29}, as well as novel antagonistic AP-1 TF-pairs (**Fig. 1f**).*”

We have also updated the figure caption for Fig. 1d-f as follows (lines 970-974):

“(d-f) Results of pY1H screen between NF- κ B and AP-1 TF-pairs and cytokine gene promoters. (d) Main network shows connections between TF-pairs and cytokine promoters. (e, f) Insets show cooperative and antagonistic relationships between NF- κ B (e) and AP-1 (f) TFs. Node size indicates the number of binding events for that TF. Edge width represents the number of cooperative or antagonistic events involving a specific TF-pair.”

1e - why is this presented in percentage instead of total count? What interactions is this summarizing? Plotting this as a bar chart (and not stacked) will make this information more clear.

In this figure, we are comparing literature overlap between all cooperative binding events, cooperative events found using the 1-AD assay design, and cooperative events found using the 2-AD design. For each of these three subsets of interactions, we wanted to display the *percentage* of interactions with literature support to show that the percentage was similar between each of the three subsets without confusion due to differences in the number of interactions assessed. We have clarified that the numbers 40, 22, and 25 above the bars represent the number of interactions included in each bar by adding the following text to the caption for this panel, which is now Figure 1g (lines 975-976): “*Numbers above each bar reflect the number of binding events assessed in each category.*”

Figure 2:

2e (and associated text) - is it expected for there to be so little overlap between the eY1H and pY1H?

Minimal overlap between eY1H and pY1H interactions is to be expected here, and, in fact, demonstrates the utility of pY1H. In this figure, we are representing PDIs between individual TFs and cytokine promoters that we derived from our cooperative binding events by pY1H. Given that in our screen cooperative binding events were considered when individual TFs have little or no binding on their own, we have excluded strong interactions by individual TFs. eY1H detects only independent binding events by individual TFs, and cannot detect binding events that require cooperativity between different TFs. Therefore, as expected, eY1H and pY1H detected

distinct sets of interactions. This demonstrates that there are many PDIs that require cooperativity and are therefore only detected by pY1H assays.

We added the following to the Results section (lines 176-179) for explanation:

“Overlap between cooperative binding-derived PDIs and eY1H interactions is minimal, as eY1H cannot detect interactions that require cooperative binding and we excluded any independent binding events by individual TFs from our pY1H analysis.”

2f is very unclear to me. What are the error bars? Is this test biased due to the nature of selection of genes you’ve screened for pY1H - intensively studied pathways/genes that have known interactors? Are the number of interactions and targets for eY1H comparable? If the eY1H data encompasses genes from less studied pathways or a much larger number of interactions, this chart is a misrepresentation of how well these assays can be validated in vivo. Removing this chart and in the text stating the percentage (and total number) of pY1H PDIs found with literature support is the more straightforward way to present this data.

We apologize for the confusion. Our pY1H and eY1H assays focused on a subset of 18 DNA baits corresponding to cytokine promoters. These 18 baits were selected because they have each been shown to be regulated by one NF- κ B and one AP-1 TF. However, the results analyzed here contain interactions from our full array of 297 TF-pairs, only 27 of which are NF- κ B or AP-1 pairs. We therefore do not expect that these 18 baits were significantly more biased for previously reported interactions with our full TF-pair array, and we consider it fair to compare these eY1H and pY1H interactions.

Furthermore, we believe it is important to demonstrate that our pY1H interactions validate at a similar or higher rate than interactions detected by standard eY1H. This suggests that PDIs that require cooperativity are equally as relevant as independent PDIs. We observed that the pY1H-derived PDIs showed a greater overlap with the literature than eY1H PDIs (14% vs 6%). To determine whether this difference was significant, we conducted a two-tailed proportion comparison test and found that the proportion of pY1H-derived PDIs with literature evidence was significantly greater than the proportion of eY1H-derived PDIs with literature evidence ($p=0.0024$). Error bars represent the standard error of proportion. We have added the following underlined text to the caption for Figure 2f (lines 984-986): *“(f) Percentage of eY1H ($n=270$) and pY1H ($n=256$) PDIs with literature evidence. Significance by two-tailed proportion comparison test. Error bars represent the standard error of proportion.”*

We agree that it would be helpful to explicitly state the percentage of literature overlap for eY1H- and pY1H-derived PDIs in the results text. Additionally, in response to reviewer #2, we have added a comparison of the degree to which eY1H- and pY1H-derived PDIs correspond to reported ChIP-seq peaks. In the text, we have added the following underlined text in the Results section (lines 179-183): *“More importantly, when compared to eY1H PDIs, pY1H-derived PDIs showed a greater overlap with the literature ($\sim 6\%$ vs. $\sim 14\%$ overlap, $p=0.0024$ by two-tailed*

proportion comparison test) and with available ChIP-seq peaks (~38% vs ~57% overlap, $p=9.7 \times 10^{-5}$ by two-tailed proportion comparison test) (Fig. 2f,g), demonstrating that pY1H assays can recover known PDIs not detectable by eY1H assays.”

Figure 3

3e - is there a way to summarize this data in a more interpretable manner? Maybe a density plot? That data is too overlapping for this to be clearly read.

As per the reviewer’s suggestion, we tried grouping TF-pairs into bins based on the tissue specificity of each TF but felt that we were losing too much information by doing so. Instead, we generated a new scatter plot making the data points small and uniform in size so each point could be better distinguished. We also adjusted the scales of the axes to maximize the graph space. Rather than using the size of each data point to indicate the expression similarity of the two TFs, we added an additional panel (new Figure 3f) which plots the difference in tissue specificity between the two TFs against the Simpson similarity of expression between the TFs. Here, the reader can see that even TF-pairs with very different tissue specificity scores are expressed in similar sets of tissues. We added the following reference to the new Fig. 3f in the Results section (lines 217-221):

“We observed that these functional relationships often occur between ubiquitous-ubiquitous and ubiquitous-specific TF-pairs (Fig. 3e). Even for ubiquitous-specific TF-pairs, TFs were expressed in overlapping sets of tissues, with 97% of all TF-pairs coexpressed in at least one tissue or cell type (Fig. 3f), indicating potential venues for cooperative and antagonistic interactions to occur in vivo.”

We have added the following to the caption for the new Figure 3f (lines 1009-1010):: “(f) Scatter plot showing the Simpson co-expression similarity and the difference in TCESS for each TF-pair showing cooperativity, antagonism, or both (complex).”

We have renumbered the previous Figures 3f and 3g to Figures 3g and 3h, respectively.

Figure 4

4a - the light green/white text combo is difficult to read.

We agree that the node and label color scheme in this panel is difficult to read. We have changed the node colors to light/pastel shades, have changed all the label text to black, and have enlarged the nodes so the labels are easier to read.

4b - this panel is too much to read and not labeled sufficiently. I recommend using 1 or 2 of these plots, moving the rest to supplement, and labeling much more. Not knowing the baits removes a lot of important information from this figure. Also, the arrows are almost impossible to see.

We agree that this panel may be difficult to read. However, we feel that all TF-pairs in this panel are important to display, so we have not moved any parts of the figure to the supplement. Instead, we have enlarged the plots so they can be more easily interpreted and added labels for the promoter baits represented.

Reviewer #2

Revision of NCOMMS-23-18312-T

The manuscript by Berenson et al. introduces paired yeast-one hybrid (pY1H), an experimental technique to screen transcription factor (TF) interactions with DNA in a high-throughput manner and assess potential mechanistic processes linked to biomolecular interactions. This assay is of major relevance for the Systems Biology / Functional Genomics community, given the need of automating the screening of TF pairs in a controlled system and allowing the genetic permutation of hundreds of TFs and target sequences.

To the best of my knowledge (Computational Biology and Biochemistry), this setup is a relevant, major improvement to the previous eY1H approaches previously developed by some of the same authors, and allows screening and mechanistic assessment of biological interactions between TF pairs e.g. competition/cooperativity/sequestering TFs. Overall, the main sections of this manuscript are very well written and presented, which makes the main work easy to read, understandable, and relevant to a broad audience.

I am listing below revisions, mostly related to data analyses, limitations, and data/code release, which in my opinion would strengthen the interpretation of this work and the accessibility of experimental data generated. I am supportive of the publication of this work, once these points are assessed for revision and reviewed,

We thank the reviewer for the positive comments and for the great suggestions to improve clarity and reproducibility.

Major revisions:

1.

(i) The distinction from observed results “competition” and “sequestration” interactions of TF1 by TF2, among others, does not seem to be the only theoretically plausible in some cases. If TF2 is not by itself able to activate the reporters, a “sequestration” annotation could be given to TF1+TF2, despite perhaps being in reality and actual a competition event, in which where TF2 by itself is faulty and unable to activate HIS3/LacZ.

(ii) As an additional observation, it looks like both 1-AD and 2-AD are always required to mechanistically interpret the observed signals.

2. These two ideas, and potentially other mechanistic limitations the authors can identify in other cases, are necessary to be further discussed in the Discussion. Some panels might require updates to contain both 1-AD and 2-AD designs, in case they are required for interpretability (e.g. 4c, S1).

We thank the reviewer for pointing out the gap in clarity. To bypass the endogenous transcriptional activity of each TF, we fuse the yeast Gal4 activation domain (AD) to one or both TFs. While it is possible that the Gal4 AD is occluded or inactive in some scenarios, it is typically known to reliably activate transcription in yeast when recruited to a promoter (Y1H and Y2H refs). To clarify, we have added the following underlined text to the Results section (lines 86-89):

“In the event of TF-DNA binding, the AD promotes the expression of both HIS3 (allowing yeast to overcome inhibition by the His3p competitive inhibitor 3-amino-1,2,4-triazole) and lacZ (producing a blue compound in the presence of X-gal), regardless of the intrinsic transcriptional activity of the TF.”

Ideally, we would test all interactions using both the 1-AD and 2-AD designs. However, this would reduce the number of pairs we can test simultaneously, and we argue that it is typically not necessary. The 2-AD design reliably detects mutual cooperative binding and sequestration, which are two key mechanisms by which TFs affect one another’s DNA occupancy. As stated in the Results section, we selected the 2-AD design for most of our screens, as we were predominantly focused on these two binding modes. We have added the following underlined text to the Discussion section to reiterate this point and acknowledge the limitations of the approach (lines 336-340):

“pY1H assays can be used for diverse applications, leveraging both the 1-AD and the 2-AD designs. While the 1-AD design can be used to distinguish between a greater number of distinct binding modes and is likely to capture more dependent binding events, the 2-AD design efficiently detects mutual cooperativity and sequestration, two key mechanisms by which TFs affect one another’s DNA occupancy.”

Fig 2g: (i) The analysis of interactions with CHIP-seq seems descriptive, and a statistical test could be required.

We agree that a statistical test is necessary to determine significant overlap between pY1H interactions and CHIP-seq data. We had included in Supplementary Figure 6f-g a statistical analysis comparing our CHIP-seq overlap to overlap with a randomized pY1H network, determining significant overlap for our cooperative binding events. We have now moved the figures for these randomization analyses to the main Figure 2h and 2i so the reader can easily see this important analysis.

We have added the following to the caption for Figure 2h and 2i (lines 990-993):

“Overlap between pY1H results and ChIP-seq peaks was compared to distributions of overlap for 10,000 randomized pY1H networks. Two-tailed statistical significance was calculated from Z-score values assuming normal distribution for overlap with the randomized networks.”

(ii) In the previous panel (2f) a comparison/enrichment comparison with eY1H was performed. Can the authors provide such a comparison and interpretation of whether the agreements of pY1H with ChIP-seq are higher than eY1H? As an additional dataset, authors could also utilize TF pairs reported by Jolma et al. (CAP-SELEX, Nature 2015), to study whether interfamily TF1-TF2-DNA complexes are also more strongly supported than eY1H.

We thank the reviewer for the suggestion. We agree that, in addition to comparing overall literature overlap between eY1H and pY1H PDIs, it would also be helpful to compare ChIP-seq overlap for these PDIs. We have completed this analysis to determine the number of pY1H- and eY1H-derived PDIs with ChIP-seq evidence, and compared this to overlap with randomized versions of the pY1H and eY1H networks. We observed that 37.5% of eY1H PDIs and 57.1% of pY1H-derived PDIs with available data had ChIP-seq evidence, a difference which was significant when analyzed using a proportion comparison test.. We have added this comparison as the new Figure 2g, a chart similar to Figure 2f. Previous Figure 2g has been renumbered to Figure 2h. We have added the following underlined text to the Results (lines 179-183):

“More importantly, when compared to eY1H PDIs, pY1H-derived PDIs showed a greater overlap with the literature (~6% vs. ~14% overlap, $p=0.0024$ by two-tailed proportion comparison test) and with available ChIP-seq peaks (~38% vs ~57% overlap, $p=9.7 \times 10^{-5}$ by two-tailed proportion comparison test) (Fig. 2f,g), demonstrating that pY1H assays can recover known PDIs not detectable by eY1H assays.”

We have added the following to the caption for Figure 2 (lines 986-988): *“(g) Percentage of eY1H ($n=176$) and pY1H ($n=226$) PDIs with ChIP-seq evidence. Significance by two-tailed proportion comparison test. Error bars represent the standard error of proportion.”*

We have also added the following to the Methods section (lines 581-592):

“Comparing eY1H and pY1H ChIP-seq overlap. The eY1H dataset consisted of 270 TF-promoter pairs, while the pY1H dataset contained 256 pairs derived from this study. We again utilized the GTRD database to obtain ChIP-seq data for ey1H dataset (See "Overlap between ChIP-seq and pY1H interactions" for more details).

Subsequently, we excluded TF-promoter pairs for which ChIP-seq information was not available. To compare the proportion of TF-promoter pairs with ChIP evidence between eY1H and pY1H, we employed a two-tailed proportion comparison test. We also performed a network randomization analysis separately for eY1H and pY1H datasets. For each dataset, we generated 10,000 networks and performed 20,000 edge-switches to assess the significance of the observed results (See: "Network randomization analysis"). Based on the 10,000 random networks generated, a Z distribution was used to obtain a Z-scores and two-tailed p-values for the original eY1H and pY1H networks."

As per the reviewer's suggestion, we looked into the overlap between our pY1H interactions and the data presented by Jolma et al. (PMID: 26550823). While Jolma et al. tested 315 TF-pairs and we tested 297 TF-pairs, only one TF-pair is common to both lists, so we were unable to complete the recommended comparison.

3.

It is my impression that pY1H datasets, and the complementary eY1H datasets, are very interesting for the ML/Genomics community, for the purpose of predicting signals based on TF sequence, DNA sequence, and interactions with other TFs, using Deep Learning models e.g. Graph-Neural networks with sequence representations. The current Supplementary Table format is generic for publication, yet it seems that a processed dataset with normalized signals per replicate, plus harmonized annotations, could allow other researchers to use this data and try to model observations using mechanistic and/or generative approaches.

We thank the reviewer for recognizing the value of the dataset and for the suggestions to make it more accessible and useful to the community. To address this, we have updated Supplementary Data 8 where we now also provide intensity and area values of the colonies corresponding to TF1-TF2, TF1-empty, and empty-TF2 yeast strains.

4.

Line 154: "Overall, this screen-detected novel instances of sequence-specific cooperativity and antagonism...". I think the interpretation of sequence-specific and/or TF-specificity effects in this work is limited. I would either replace "sequence-specific" statements with "gene interactions" whenever suitable. Alternatively, I think it could be better to exemplify how these interactions are suggested to happen at the DNA sequence level e.g. are promoters with strong/weak bZIP motifs still preferred for cooperative interactions? Are cooperative/antagonistic promoters showing a higher/lower number of motifs for certain TF families? Results could also be interpreted by showing mapped motifs and specific sequence modifications. Examples for either 1d, 2d, 4d (XCL1 and TNF5F8), or 5c (any hTF-vTR) could easily generate and enhance the interpretation of results. Authors could decide on at least two examples from that list, or other suitable ones, for interpretation.

We agree that additional motif analysis would strengthen our point that relationships between TFs are sequence specific. To illustrate this, we had previously analyzed motifs for the MAX-MXI1 in Supplementary Figure 7a by searching the DNA bait sequences for the MAX and MXI1 DNA binding motifs (see Methods: Identification of binding sites of TF-pairs in cytokine

promoters). We observed two overlapping MAX/MXI1 dual motifs in the bait at which MAX and MXI1 bound cooperatively in pY1H assays (CCL5), but no dual motifs, close motif pairs, or individual MXI1 motifs in the baits at which MXI1 antagonized MAX (IL18, CCL15). We have updated the schematic for this motif analysis in what is now Supplementary Fig. 8a.

Further analysis of DNA binding motifs or other determinants of cooperative and antagonistic binding is limited by a number of factors. The preferred motif grammar for many TF-pairs is unknown, and existing heterodimer-DNA binding motifs are difficult to compare to one another or to homodimer binding motifs, as they are often derived from separate experiments which rely on TF-specific antibodies. Therefore, relative binding affinities cannot be predicted from motif logos.

Motifs cannot be determined from the pY1H assays analyzed in this study due to the small number of DNA bait sequences tested and the length of each bait (~2kb). Furthermore, we are still actively investigating how chromatin context in our yeast strains may contribute to eY1H and pY1H results, which we hope to explore by generating larger datasets.

From a biological perspective, we suspect that cooperative and antagonistic events may be the result of lower affinity TF-DNA interactions which are missed by standard motif analysis. For example, cooperative binding may occur at DNA regions carrying weak or noncanonical binding sites for both TFs. While more extensive analysis is beyond the scope of this manuscript, this is a fascinating area of study which we hope to explore in future manuscripts.

When mentioning specific examples in the text which we did not explore using motif analysis, we have replaced “sequence-specific” with “bait-specific” or “DNA region-specific” to avoid overselling our conclusions. These include the following:

In the Abstract, lines 23-24: *“We provide evidence that a wide variety of TFs are subject to modulation by other TFs in a DNA region-specific manner.”*

In the Introduction, lines 70-71: *“This approach reveals that these functional relationships occur across well-known and lesser-known TF-pairs in a DNA region-specific manner.”*

In the Results section:

Lines 128-131: *“Interestingly, though sequestration is generally expected to cause global loss of binding of the sequestered TF, some sequestering relationships such as that between REL and RELB were DNA bait-specific, as RELB did not prevent REL binding at all promoters tested.”*

Lines 148-150: *“Overall, this screen detected additional instances of DNA bait-specific cooperativity and antagonism between highly-studied NF-κB and AP-1 TFs.”*

We also changed a subheading in the Results section (lines 195-196) from *“TF-TF relationships are sequence-specific and connect ubiquitous and tissue-specific TFs”* to *“TF-TF relationships are DNA region-specific and connect ubiquitous and tissue-specific TFs.”*

5.

LN308: “STAT3 and STAT1 affect the equilibrium between STAT3/STAT3 homodimers” Here it would be nice to clarify if the authors interpret that these complex events are happening once either STAT3 or STAT1 are binding to DNA, or in a DNA-independent manner. A protein structure visualization of those cases, highlighting activation domains and/or deleted regions, or the general scheme of this case as a supplementary figure, could be useful for the interpretation of this point.

We thank the reviewer for this suggestion. Using AlphaFold 2, we generated predicted structures for STAT3 and the STAT1 isoforms of interest (STAT1-202 and STAT1-201). We then modeled dimerization of STAT3 with each STAT1 isoform to determine whether the STAT1-201 isoform could affect STAT1-STAT3 dimerization or DNA binding. We observed that the additional C-terminal region in the STAT1-201 isoform would not interfere with STAT1-STAT3 dimerization in the antiparallel conformation (where the C-terminal domains are distal from the site of dimerization), but could interfere with dimerization in the parallel conformation, which is the primary conformation for DNA binding (PMID: 17216035, 30267440). This supports an antagonistic mechanism by which STAT1-201 dimerizes with STAT3, decreases the number of STAT3 subunits available to form STAT3-STAT3 homodimers, and forms a STAT1-STAT3 dimer that is unable to bind DNA.

We have included visualizations of these predicted structures as the new Supplementary Figure 9, and added the following to the Results text (lines 270-279):

*“However, STAT3 binding is also antagonized by the STAT1-201 isoform, which retains its DNA binding domain but has an additional C-terminal domain. To determine the potential mechanism of antagonism, we used AlphaFold 2 to predict structures of dimers between STAT3 and the STAT1-202 and STAT1-201 isoforms. We observed that the additional C-terminal region in STAT1-201 likely does not interfere with STAT1-STAT3 dimerization in the antiparallel conformation (where the C-terminal domains are distal from the site of dimerization), but could interfere with dimerization in the parallel conformation, which is the primary conformation for DNA binding^{40,41}(**Supplementary Fig. 9**). This supports an antagonistic mechanism by which STAT1-201 dimerizes with STAT3, decreases the number of STAT3 subunits available to form STAT3-STAT3 homodimers, and forms a STAT1-STAT3 dimer that is unable to bind DNA.”*

We have also added the following subsection to the Methods (lines 739-744):

“Structural predictions of STAT1/STAT3 dimers. We utilized AlphaFold 2 to generate the structures of STAT3, STAT1-201, and STAT1-202, employing the following parameters: --model_preset=monomer and --db_preset=full_dbs. To visualize the structures, we utilized Pymol and selected the surface and cartoon representations. Parallel and antiparallel conformations of dimers were arranged manually in Pymol.”

6.

The provided code repository (LN568 <https://github.com/mahir1010/D.I.S.H.A> and DISHA viewer) is weak. Methods sections require rewriting and linking to the right processing notebook/scripts, with an example dataset e.g. “Image processing” As currently presented, the code does not allow reproducing the results from this work. I please request the authors to present simplified documentation notebooks (e.g. Jupyter notebooks) showcasing the most relevant data analyses conducted in this work (summary barplots, ChIP-seq, network permutations, TCESS comparisons), with a processed dataset that is retrievable, and with minimal installation dependencies. This does not require a software release and can be mounted into a reproducibility-specific GitHub repository, per figure and/or panel.

We thank the reviewer for the request and agree this will help reproduce our analyses and results. We have now uploaded all our annotated scripts to our GitHub repository (https://github.com/jfuxman/PY1H_NatComm2023) as Jupyter notebooks for Python scripts, and R markdown for R scripts, together with links to the data so that readers can run the code themselves. We have also updated the following methods sections by providing links to a specific folder associated with each analysis:

Predicting possible TF-TF interactions based on homology (lines 399-401)

Bioinformatics analysis of TF-prey sequencing data (lines 492-494)

Image processing (lines 552-553)

Comparing eY1H and pY1H ChIP-seq overlap (lines 593-598)

Overlap between ChIP-seq and pY1H interactions (lines 611-613)

Identification of binding sites of TF-pairs in cytokine promoters (lines 638-640)

Network randomization analysis (657-662)

Paralog partner similarity (lines 690-691)

TF expression analysis (lines 709-710)

Tissue/cell type expression specificity scoring of genes (lines 722-723)

Transcription factors co-expression among tissue/cell types (lines 736-737)

Minor.

LN 39: This claim makes sense, but a reference to a TF-paralog competition example is needed.

To specify the case in which TF paralogs compete with one another for similar DNA binding sites, we have added the following underlined text to the Introduction (lines 38-40):

“Other TFs antagonize one another by sequestration via protein-protein interactions or by competing for binding at specific DNA sites (e.g., paralogs that recognize the same motif³⁷).”

This includes references to PMID: 33975875 and PMID: 35404235.

LN 46: I think a reference to the futility theorem review from Wasserman’s lab is also needed here (PMID 15131651). The current reference seems to be an analysis based on that tool

We have added the recommended reference, PMID: 15131651, in line 46.

LN 46: “Predictions are generally more challenging for heterodimers” This seems to be mainly due to wet lab, experimental limitations. To my knowledge, there’s no benchmark on whether computational predictions are more challenging for hetero- than heterodimers motifs. Provide reference, if any.

We have made the following replacement in the Introduction text (lines 46-48) to clarify that the challenges in computational prediction are largely due to limitations in experimentally determined binding motifs for heterodimers: *“Predictions are generally more challenging for TF heterodimers, exacerbated by the fact that as binding motifs have not been determined for most heterodimers due to challenges in producing and purifying protein complexes in vitro.”*

LN 75: “Further expanding the TF landscape”, I think “TF interactome landscape” might be a suitable term there, unless the authors are referring to something else.

We have replaced “TF landscape” with “*TF interactome landscape*” in line 75.

LN147: Shortly clarify in text/captions how these literature interactions are gathered related to cytokine-TF. I understand it’s a single paper by Santoso et al., once getting into the references.

To compare our interactions to those previously reported in the literature, we used the CytReg database which we previously generated. This database compiles information about which TFs have been shown to bind to regulatory regions or regulate the expression of each human cytokine gene. To clarify, we have added the following underlined text to the Results section (lines 137-140): *“For 70% of these events, one or both TFs were known to bind to the regulatory regions or regulate the expression of that cytokine, as per the CytReg Database (https://cytreg.bu.edu/search_v2.html).”*

LN179: caption/text needs to indicate if the statistical test is one- or two-tailed.

The proportion comparison test we conducted was two-tailed. We have added the following underlined text to the figure caption (lines 984-986): "(f) Percentage of eY1H (n=270) and pY1H (n=256) PDIs with literature evidence. Significance by two-tailed proportion comparison test. Error bars represent the standard error of proportion." We also added the following underlined text to the Results section (lines 179-183):

"More importantly, when compared to eY1H PDIs, pY1H-derived PDIs showed a greater overlap with the literature (~6% vs. ~14% overlap, $p=0.0024$ by two-tailed proportion comparison test) and with available ChIP-seq peaks (~38% vs ~57% overlap, $p=9.7 \times 10^{-5}$ by two-tailed proportion comparison test) (Fig. 2f,g), demonstrating that pY1H assays can recover known PDIs not detectable by eY1H assays."

3C: "other pairs" instead of TF pairs.

We have fixed this legend to say "Other TF-pairs" rather than "TF-pairs."

3D: The number of observations in each violin plot would be relevant to know.

We have added the number of interactions above each violin plot in Figure 3d. We have added the following to the figure caption to explain these numbers (lines 1002-1003): "*Numbers above each column reflect the number of binding events assessed in each group.*"

2d: The individual number of coop. and antagonistic interactions are not clear in this barplot. I recommend visually separating those two, and highlighting percentages as well e.g. all bars higher or equal to one = 63%.

We agree that it is helpful to see the distributions of cooperative and antagonistic events separately to assess the contribution of each type of interaction. We had previously included a version of this figure depicting cooperative and antagonistic interactions separately in Supplementary Figure 6a-b. We believe that it is best to keep the existing graph with cooperative and antagonistic events combined as Figure 2d, as we want to show how many of our pairs participate in either of these non-independent binding modes. However, to direct readers to the supplementary cooperativity and antagonism graphs, we have added the following underlined text in the results section, specifying the percentage of TF-pairs in our

array with ≥ 1 cooperative interaction and the the percentage with ≥ 1 antagonistic interaction (lines 168-171):

“Of the TF-pairs tested, 63% showed at least one cooperative or antagonistic interaction, including 60 of the 88 TF-pairs selected based on homology (Fig. 2d). Specifically, 32% of TF-pairs showed at least one cooperative interaction and 38% of TF-pairs showed at least one antagonistic interaction (Supplementary Fig. 6).”

We have also added an indication in both Figure 2d and Supplementary Fig. 6 of the percentage of TF-pairs that showed at least one interaction in our screen. We have added the following to the captions for Figure 2d (lines 982-983) and Supplementary Figure 6: *“The percentage of TF-pairs with at least one cooperative or antagonistic event is indicated.”*

Figure 5b: It is unclear if the heatmap enumerated all interactions tested and or found. Can the authors describe in results/captions why not all vTR and hTFs interactions were tested e.g. were STAT vs. vTFs interactions tested, or no interactions were found? Highlighting with labels non-zero heatmap cells the light cells could be suitable to highlight rare cases with only “1” or “2” observations.

Figure 5b depicts all hTF-vTR pairs tested in our screen, which we selected based on known or suspected protein-protein interactions. We have added the following text to the Results section (lines 288-290):

“We generated a pY1H array of 113 protein pairs containing one human TF and one vTR that are known or suspected to interact by PPIs (Fig. 5b) and screened for interactions with 83 promoters of cancer-related genes.”

We have also added to the caption for Figure 5b (lines 1034-1036): *“hTF-vTR pairs were selected based on known PPIs between the two proteins or homology with known pairs.”*

In response to reviewer #3, we have expanded this figure so each column represents a single vTR, rather than a viral species. We have also added number labels to each non-zero cell to show the number of pairs represented by each box, as suggested.

Reviewer #3

Comments for Authors:

In the manuscript entitled "Paired yeast one hybrid assays to detect DNA-binding cooperativity and antagonism across transcription factors," the authors present not only a novel method to investigate the interplay of transcription factors (TFs) in gene regulation but also a multitude of data obtained by different settings analyzed in the study that can be used by other researchers as starting point for future studies. The abstract and the main body of the manuscript are written of overall clarity and readability and the figures are very well-designed. The method allows the investigation of transcriptional regulation of a specific promoter sequence by a defined pair of transcription factors. The findings suggesting that the role of a TF depends on its TF partner as well as on the target DNA sequence. Although no in-depth analysis was performed for a specific TF pair (e.g. TF interaction/degradation studies, in mammalian cells), the method and the results presented are of immediate interest to many people in the field of transcriptional regulation including cell biologists, oncologists, immunologists, and virologists investigating virus host interactions on the level of cellular and viral gene expression that influences virus replication and virus induced oncogenesis. Therefore, this study represents a major advance in a broad scientific field.

Key results: The study provides a novel method that is broadly applicable and data that shed light into TF cooperativity and antagonism, isoform usage and explains the importance of the cell type specific background for the understanding of the mode of action of a specific TF on a defined DNA sequence. This study addresses a critical aspect of transcriptional regulation by focusing on the cooperation and antagonism between TFs. The topic is highly relevant to a broad readership, as it contributes to our understanding of the complex mechanisms governing gene expression. The authors' novel approach represents a significant advancement in the field and opens up new avenues for studying TF interactions.

Validity of the approach: The manuscript provides a comprehensive analysis of the proposed method's efficacy by testing it on the well-established examples, AP1 and NFkB subunits, on cytokine promoter regions. This validation step is crucial to demonstrate the robustness and reliability of the pY1H assays. Furthermore, the authors perform a broad screening experiment including almost 300 TF pairs, contributing to our understanding of cooperativity and antagonism across specific transcription factors which can explain to some extent the complexity of tissue-specific gene regulation. This aspect highlights the versatility and potential applications of their method in uncovering the interplays of TF interactions across various biological contexts.

An additional strength of this manuscript lies in the authors' comparison of the effects of TF isoforms and viral factors on cellular TFs. These comparisons offer valuable insights into the functional consequences of TF variations and viral-host interactions on specific promoter sequences. By exploring these diverse scenarios, the authors provide a comprehensive evaluation of their method's capabilities and showcase its broad utility. In addition, limitations of the methods are also clearly stated by the authors. The data interpretation is robust, valid and reliable. The conclusions drawn by the authors are comprehensible.

After a careful evaluation of the manuscript, I am pleased to recommend it for publication with only minor changes.

We thank the reviewer for the very positive comments on the significance of our work and the validity of our approach, and for the suggestions to further improve the manuscript.

To improve the manuscript, I suggest addressing the following minor points:

Interestingly, pY1H cooperative events significantly overlapped with motif predictions and ChIP-seq data while antagonistic TF-pairs are less predictable. The authors explain this with sequestration rather than competitive binding of both TFs.

Here, the authors should also mention that viral factors such as HPV16 E7 are potent degraders of cellular TFs such as pRB, MYPOP or PTPN14. In their assay the authors could determine the putative degradation of TFs in yeast co-expressing viral proteins and protein quantification. Alternatively, consider expanding the discussion section to elaborate on this scenario.

We agree that it is important to consider other factors that might affect relationships between TFs. To address this, we have added the following underlined text to the Discussion section (lines 329-332):

“Therefore, orthogonal experiments may be required to determine the specific contexts in which these events occur, or whether they are affected by post-translational modifications (e.g., IRFs and STATs⁵⁰) or by one TF targeting the other for degradation (e.g., viral HPV-16 E7^{51,52}).”

Specify/label viral proteins in Figure 5 (e.g. “HPV16 E7” instead of “HPV16”). This will facilitate readers' comprehension of the figure.

We have modified Figure 5b so each column represents a single viral protein rather than collapsing by viral species. We have also added number labels to each non-zero cell to specify the number of protein pairs represented.

I also recommend naming the virus families in the abstract (HPV, EBV, AdV, HIV, HTLV, HBV) in order to strengthen the attention of the findings by virologists.

We have added mention of HPV, EBV, and HIV to the abstract by adding the following underlined text (lines 24-27): *“We also demonstrate that TF-TF relationships are often affected by alternative isoform usage, and identify cooperativity and antagonism between human TFs and viral proteins from human papillomaviruses, Epstein-Barr virus, and other viruses.”*

Line 156: “h ow”

We have fixed this typo.

In summary, the manuscript is of high interest to a broad readership. The authors' significant contributions to the understanding of transcriptional regulation, including the exploration of TF isoforms and viral factors, make this study highly impactful. With the suggested minor revisions, this manuscript will be an excellent addition to Nature Communications, advancing the field of gene regulation and inspiring further research in this area.

REVIEWERS' COMMENTS

Reviewer #1 (Remarks to the Author):

The authors have sufficiently addressed my and the other reviewers' comments. I am pleased to see this paper published and this method used in future studies.

Reviewer #2 (Remarks to the Author):

(Reviewer #2) - R02

The authors have addressed the majority of my revisions, included new panels for those, and updated the manuscript text accordingly. The author's explanations for two specific major comments are also satisfactory e.g. technical limitations of their approach (Major 1) and the requested motif analysis being out of scope (Major 4).

Regarding Major 6, I checked their code and the most relevant analyses can be inspected. If relevant, I imagine GitHub issues will be submitted for verification. A suggestion to the authors is to number chronologically their analyses e.g. "01_Identification of binding sites of TF-pairs in cytokine promoters". This would help follow the manuscript analyses sequentially. Finally, the hyperlinks in the merged PDF are not opening in my PDF reader, due to the link covering two lines and only parsing half of it. These points are overall minor, and I am sure can be addressed during post-revision edits.

I recommend for publication. Thank you,

Reviewer #3 (Remarks to the Author):

All my points have been addressed. Therefore I propose to publish the manuscript as it is.

RESPONSE TO REVIEWERS' COMMENTS

Reviewer #1 (Remarks to the Author):

The authors have sufficiently addressed my and the other reviewers' comments. I am pleased to see this paper published and this method used in future studies.

We thank the reviewer for their time and helpful suggestions.

Reviewer #2 (Remarks to the Author):

(Reviewer #2) - R02

The authors have addressed the majority of my revisions, included new panels for those, and updated the manuscript text accordingly. The author's explanations for two specific major comments are also satisfactory e.g. technical limitations of their approach (Major 1) and the requested motif analysis being out of scope (Major 4).

Regarding Major 6, I checked their code and the most relevant analyses can be inspected. If relevant, I imagine GitHub issues will be submitted for verification. A suggestion to the authors is to number chronologically their analyses e.g. "01_Identification of binding sites of TF-pairs in cytokine promoters". This would help follow the manuscript analyses sequentially. Finally, the hyperlinks in the merged PDF are not opening in my PDF reader, due to the link covering two lines and only parsing half of it. These points are overall minor, and I am sure can be addressed during post-revision edits.

I recommend for publication. Thank you,

We thank the reviewer for their suggestions. We have considered various approaches to make our code as accessible as possible for those hoping to recreate our analyses. As suggested by the reviewer, we did consider numbering the GitHub analyses chronologically according to the order in which they are mentioned in the paper. However, we felt that it was more helpful to keep the analyses organized according to the methods section in which they are described, as a single methods section may correspond to an analysis conducted at multiple points throughout the paper (e.g., network randomization analysis). We have

therefore decided to retain the format initially suggested by the reviewer, in which each methods section is accompanied by the direct GitHub link to the corresponding code.

We have ensured that hyperlinks in the manuscript file are now functional.

Reviewer #3 (Remarks to the Author):

All my points have been addressed. Therefore I propose to publish the manuscript as it is.

We thank the reviewer for their time and helpful suggestions.